# SALSA: Attacking Lattice Cryptography with Transformers

**Emily Wenger**[*]
University of Chicago

**Mingjie Chen**[*]
University of Birmingham

**Francois Charton**[†]
Meta AI

**Kristin Lauter**[†]
Meta AI

## Abstract

Currently deployed public-key cryptosystems will be vulnerable to attacks by full-scale quantum computers. Consequently, "quantum resistant" cryptosystems are in high demand, and lattice-based cryptosystems, based on a hard problem known as Learning With Errors (LWE), have emerged as strong contenders for standardization. In this work, we train transformers to perform modular arithmetic and mix half-trained models with statistical cryptanalysis techniques to propose SALSA: a machine learning attack on LWE-based cryptographic schemes. SALSA can fully recover secrets for small-to-mid size LWE instances with sparse binary secrets, and may scale to attack real-world LWE-based cryptosystems.

## 1 Introduction

The looming threat of quantum computers has upended the field of cryptography. Public-key cryptographic systems have at their heart a difficult-to-solve math problem that guarantees their security. The security of most current systems (e.g. [58, 28, 50]) relies on problems such as integer factorization, or the discrete logarithm problem in an abelian group. Unfortunately, these problems are vulnerable to polynomial time quantum attacks on large-scale quantum computers due to Shor's Algorithm [62]. Therefore, the race is on to find new post-quantum cryptosystems (PQC) built upon alternative hard math problems.

Several leading candidates in the final round of the 5-year NIST PQC competition are *lattice-based cryptosystems*, based on the hardness of the Shortest Vector Problem (SVP) [2], which involves finding short vectors in high dimensional lattices. Many cryptosystems have been proposed based on hard problems which reduce to some version of the SVP, and known attacks are largely based on lattice-basis reduction algorithms which aim to find short vectors via algebraic techniques. The LLL algorithm [43] was the original template for lattice reduction, and although it runs in polynomial time (in the dimension of the lattice), it returns an exponentially bad approximation to the shortest vector. It is an active area of research [22, 48, 5] to fully understand the behavior and running time of a wide range of lattice-basis reduction algorithms, but the best known classical attacks on the PQC candidates run in time exponential in the dimension of the lattice.

In this paper, we focus on one of the most widely used lattice-based hardness assumptions: Learning With Errors (LWE) [56]. Given a dimension $n$, an integer modulus $q$, and a secret vector $\mathbf{s} \in \mathbb{Z}_q^n$, the learning with errors problem is to find the secret given noisy inner products with random vectors. LWE-based encryption schemes encrypt a message by *blinding* it with a noisy inner product. Given a random vector $\mathbf{a} \in \mathbb{Z}_q^n$, the noisy inner product is $b := \mathbf{a} \cdot \mathbf{s} + \mathbf{e} \mod q$, where $\mathbf{e}$ is an "error" vector sampled from a narrow Gaussian distribution (so its entries are small, thus the reference to noise).

---

[*]Co-first authors, work done while at Meta AI

[†]Co-last authors

36th Conference on Neural Information Processing Systems (NeurIPS 2022).

Interestingly, the assumption for cryptographic applications is that the Learning With Errors problem is hard: given a lot of noisy inner products of random vectors with a secret vector, it should be hard to learn the secret vector. However, in Machine Learning we make the opposite assumption: given a lot of noisy data, we can still learn patterns from it. So in this paper we investigate the possibility to train ML models to learn from LWE samples.

To that end, we propose **SALSA**, a technique for performing **S**ecret-recovery **A**ttacks on **LWE** via **S**equence to sequence models with **A**ttention. SALSA trains a language model to predict $b$ from $\mathbf{a}$, and we develop two algorithms to recover the secret vector $\mathbf{s}$ using this trained model.

Our paper has three main **contributions**. We demonstrate that **transformers can perform modular arithmetic** on integers and vectors. We show that transformers trained on LWE samples can be used to distinguish LWE instances from random. This can be further turned into **two algorithms that recover binary secrets**. We show how these techniques yield a practical attack on LWE based cryptosystems and demonstrate its efficacy in the **cryptanalysis of small and mid-size LWE instances with sparse binary secrets**. Our code is available at `https://github.com/facebookresearch/SALSA`.

## 2 Lattice Cryptography and LWE

### 2.1 Lattices and Hard Lattice Problems

An integer lattice of dimension $n$ over $\mathbb{Z}$ is the set of all integer linear combinations of $n$ linearly independent vectors in $\mathbb{Z}^n$. In other words, given $n$ such vectors $\mathbf{v}_i \in \mathbb{Z}^n, i \in \mathbb{N}_n$, we define the lattice $\Lambda(\mathbf{v}_1, ..\mathbf{v}_n) := \{\sum_{i=1}^n a_i \mathbf{v}_i \mid a_i \in \mathbb{Z}\}$. Given a lattice $\Lambda$, the Shortest Vector Prob-

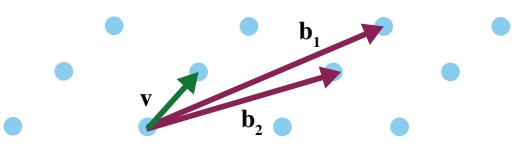

Figure 1: The dots form a lattice $\Lambda$, generated by vectors $\mathbf{b}_1$, $\mathbf{b}_2$. $\mathbf{v}$ is the shortest vector in $\Lambda$.

lem (SVP) asks for a nonzero vector $\mathbf{v} \in \Lambda$ with minimal norm. Figure 1 depicts a solution to this problem in the trivial case of a 2-dimensional lattice, where $\mathbf{b}_1$ and $\mathbf{b}_2$ generate a lattice $\Lambda$ and the green vector is the shortest vector in $\Lambda$.

The best known algorithms to find exact solutions to SVP take exponential time and space with respect to $n$, the dimension of the lattice [49]. There exist lattice reduction algorithms to find approximate shortest vectors, such as LLL [43] (polynomial time, but exponentially bad approximation), or BKZ [22]. The shortest vector problem and its approximate variants are the hard mathematical problems that serve as the core of lattice-based cryptography.

### 2.2 LWE

The Learning With Errors (LWE) problem, introduced in [56], is parameterized by a dimension $n$, the number of samples $m$, a modulus $q$ and an error distribution $\chi$ (e.g., the discrete Gaussian distribution) over $\mathbb{Z}_q = \{0, 1, \ldots, q-1\}$. Regev showed that LWE is at least as hard as quantumly solving certain hard lattice problems. Later [53, 46, 14], showed LWE to be classically as hard as standard worst-case lattice problems, therefore establishing a solid foundation for building cryptographic schemes on it.

**LWE and RLWE.** The LWE distribution $\mathcal{A}_{\mathbf{s},\chi}$ consists of pairs $(\mathbf{A}, \mathbf{b}) \in \mathbb{Z}_q^{m \times n} \times \mathbb{Z}_q^n$, where $\mathbf{A}$ is a uniformly random matrix in $\mathbb{Z}_q^{m \times n}$, $\mathbf{b} = \mathbf{A}\mathbf{s} + \mathbf{e} \bmod q$, where $\mathbf{s} \in \mathbb{Z}_q^n$ is the secret vector sampled uniformly at random and $\mathbf{e} \in \mathbb{Z}_q^m$ is the error vector sampled from the error distribution $\chi$. We call the pair $(\mathbf{A}, \mathbf{b})$ an LWE sample, yielding $n$ LWE instances: one row of $\mathbf{A}$ together with the corresponding entry in $\mathbf{b}$ is one *LWE instance*. There is also a ring version of LWE, known as the Ring Learning with Errors (RLWE) problem (described further in Appendix A.1).

**Search-LWE and Decision-LWE.** We now state the LWE hard problems. The search-LWE problem is to find the secret vector $\mathbf{s}$ given $(\mathbf{A}, \mathbf{b})$ from $\mathcal{A}_{\mathbf{s},\chi}$. The decision-LWE problem is to distinguish $\mathcal{A}_{\mathbf{s},\chi}$ from the uniform distribution $\{(\mathbf{A}, \mathbf{b}) \in \mathbb{Z}_q^{m \times n} \times \mathbb{Z}_q^n \colon \mathbf{A} \text{ and } \mathbf{b} \text{ are chosen uniformly at random})\}$. [56] provided a reduction from search-LWE to decision-LWE . We give a detailed proof of this reduction in Appendix A.2 for the case when the secret vector $\mathbf{s}$ is binary (i.e. entries are 0 and 1). In Section 4.3, our Distinguisher Secret Recovery method is built on this reduction proof.

**(Sparse) Binary secrets.** In LWE based schemes, the secret key vector $\mathbf{s}$ can be sampled from various distributions. For efficiency reasons, binary distributions (sampling in $\{0, 1\}^n$) and ternary distributions (sampling in $\{-1, 0, 1\}^n$) are commonly used, especially in homomorphic encryption [4]. In fact, many implementations use a sparse secret with Hamming weight $h$ (the number of 1's

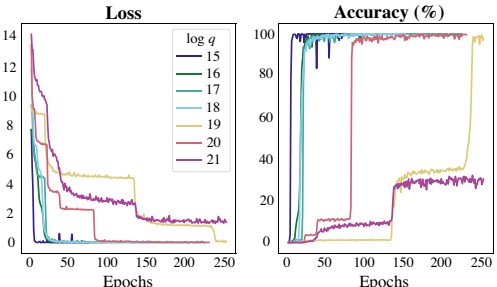

**Loss** | **Accuracy (%)**

Figure 2: **Learning modular multiplication for various moduli** $q$. Test loss and accuracy for $q$ with $\lceil \log_2(q) \rceil$ from 15 to 21. 300,000 training examples per epoch. One layer transformers with 512 dimensions, 8 attention heads, integers encoded in base 81.

Table 1: **Size of the training sets required for learning modular inversion.** Base-2 logarithm of the number of examples needed to reach 95% accuracy, for different values of $\lceil \log_2(q) \rceil$ and bases. '-' means 95% accuracy not attained after 90 million examples.

| $\lceil \log_2(q) \rceil$ | Base | | | | | | | | |
|---|---|---|---|---|---|---|---|---|---|
| | 2 | 3 | 5 | 7 | 24 | 27 | 30 | **81** | 128 |
| 15 | 23 | 21 | 23 | 22 | 20 | 23 | 22 | **20** | 20 |
| 16 | 24 | 22 | 22 | 22 | 22 | 22 | 22 | **22** | 21 |
| 17 | - | 23 | 25 | 22 | 23 | 24 | 22 | **22** | 22 |
| 18 | - | 23 | 25 | 23 | 23 | 24 | 25 | **22** | 22 |
| 19 | - | 23 | - | 25 | 25 | 24 | - | **25** | 24 |
| 20 | - | - | - | - | 24 | 25 | 24 | **24** | 25 |
| 21 | - | 24 | - | 25 | - | - | - | - | 25 |
| 22 | - | - | - | - | - | 25 | - | - | 25 |
| 23 | - | - | - | - | - | - | - | - | - |

in the binary secret). For instance, HEAAN uses $n = 2^{15}$, $q = 2^{628}$, ternary secret and Hamming weight 64 [23]. For more on the use of sparse binary secrets in LWE, see [6, 25].

## 3 Modular Arithmetic with Transformers

Two key factors make breaking LWE difficult: the presence of error and the use of modular arithmetic. Machine learning (ML) models tend to be robust to noise in their training data. In the absence of a modulus, recovering $\mathbf{s}$ from observations of $\mathbf{a}$ and $b = \mathbf{a} \cdot \mathbf{s} + e$ merely requires linear regression, an easy task for ML. Once a modulus is introduced, attacking LWE requires performing linear regression on an n-dimensional torus, a much harder problem.

Modular arithmetic therefore appears to be a significant challenge for an ML-based attack on LWE. Previous research has concluded that modular arithmetic is difficult for ML models [52], and that transformers struggle with basic arithmetic [51]. However, [17] showed that transformers can compute matrix-vector products, the basic operation in LWE, with high accuracy. As a first step towards attacking LWE, we investigate whether these results can be extended to the modular case.

We begin with the one-dimensional case, training models to predict $b = as \mod q$ from $a$ for some fixed unknown value of $s$ when $as \in \mathbb{Z}_q$. This is a form of modular inversion since the model must implicitly learn the secret $s$ in order to predict the correct output $b$. We then investigate the $n$-dimensional case, with $\mathbf{a} \in \mathbb{Z}_q^n$ and $\mathbf{s}$ either in $\mathbb{Z}_q^n$ or in $\{0,1\}^n$ (binary secret). In the binary case, this becomes a (modular) subset sum problem.

### 3.1 Methods

**Data Generation.** We generate training data by fixing the modulus $q$ (a prime with $15 \leq \lceil \log_2(q) \rceil \leq 30$, see the Appendix B), the dimension $n$, and the secret $\mathbf{s} \in \mathbb{Z}_q^n$ (or $\{0,1\}^n$ in the binary case). We then sample $\mathbf{a}$ uniformly in $\mathbb{Z}_q^n$ and compute $b = \mathbf{a} \cdot \mathbf{s} \mod q$, to create data pair $(a, b)$.

**Encoding.** Integers are encoded in base B (usually, B=81), as a sequence of digits in $\{0, \ldots B-1\}$. For instance, $(a, b) = (16, 3)$ is represented as the sequences [1,0,0,0,0] and [1,1] in base 2, or [2,2] and [3] in base 7. In the multi-dimensional case, a special token separates the $\mathbf{a}$ coordinates.

**Model Training.** The model is trained to predict $b$ from $\mathbf{a}$, for an unknown but fixed value of $\mathbf{s}$. We use sequence-to-sequence transformers [67] with one layer in the encoder and decoder, 512 dimensions and 8 attention heads. We minimize a cross-entropy loss, and use the Adam optimizer [39] with a learning rate of $5 \times 10^{-5}$. At epoch end (300000 examples), model accuracy is evaluated over a test set of 10000 examples. We train until test accuracy is 95% or loss plateaus for 60 epochs.

### 3.2 Results

**One-Dimensional.** For a fixed secret $s$, modular multiplication is a function from $\mathbb{Z}_q$ into itself, that can be learned by memorizing $q$ values. Our models learn modular multiplication with high accuracy for values of $q$ such that $\lceil \log_2(q) \rceil \leq 22$. Figure 2 presents learning curves for different values of $log_2(q)$. The loss and accuracy curves have a characteristic step shape, observed in many of our experiments, which suggests that "easier cases" (small values of $\lfloor as/q \rfloor$) are learned first.

The speed of learning and the training set size needed to reach high accuracy depend on the problem difficulty, i.e. the value of $q$. Table 1 presents the $\lceil \log_2 \rceil$ of the number of examples needed to reach $95\%$ accuracy for different values of $\lceil \log_2(q) \rceil$ and base $B$. Since transformers learn from scratch, without prior knowledge of numbers and moduli, this procedure is not data-efficient. The number of examples needed to learn modular multiplication is between $10q$ and $50q$. Yet, these experiments prove that transformers can solve the modular inversion problem in prime fields.

Table 1 illustrates an interesting point: learning difficulty depends on the base used to represent integers. For instance, base 2 and 5 allow the model to learn up to $\lceil \log_2(q) \rceil = 17$ and 18, whereas base 3 and 7 can reach $\lceil \log_2(q) \rceil = 21$. Larger bases, especially powers of small primes, enable faster learning. The relation between representation base and learning difficulty is difficult to explain from a number theoretic standpoint. Additional experiments are in Appendix B.

**Multidimensional integer secrets.** In the $n$-dimensional case, the model must learn the modular dot product between vectors $\mathbf{a}$ and $\mathbf{s}$ in $\mathbb{Z}_n$. The proves to be a much harder problem. For $n = 2$, with the same settings, small values of $q$ (251, 367 and 967) can be learned with over $90\%$ accuracy, and $q = 1471$ with $30\%$. In larger dimension, all models fail to learn. Increasing model depth to 2 or 4 layers, or dimension to 1024 or 2048 and attention heads to 12 and 16, improves data efficiency (less training samples are needed), but does not scale to larger values of $q$ or $n > 2$.

**Multidimensional binary secrets.** Binary secrets make $n$-dimensional problems easier to learn. For $n = 4$, our models solve problems with $\lceil \log_2(q) \rceil \leq 29$ with more than $99.5\%$ accuracy. For $n = 6$ and 8, we solve cases $\lceil \log_2(q) \rceil \leq 22$ with more than $85\%$ accuracy. But we did not achieve high accuracy for larger values of $n$. So in the next section, we introduce techniques for recovering secrets from a partially trained transformer. We then show that these additional techniques allow recovery of sparse binary secrets for LWE instances with $30 \leq n \leq 128$ (so far).

## 4    Introducing SALSA: LWE Cryptanalysis with Transformers

Having established that transformers can perform integer modular arithmetic, we leverage this result to propose SALSA, a method for **S**ecret-recovery **A**ttacks on **L**WE via **S**eq2Seq models with **A**ttention.

### 4.1    SALSA Ingredients

SALSA has three modules: a **transformer model** $\mathcal{M}$, a **secret recovery algorithm**, and a **secret verification procedure**. We assume that SALSA has access to a number of LWE instances in dimension $n$ that use the same secret, i.e. pairs $(\mathbf{a}, b)$ such that $b = \mathbf{a} \cdot \mathbf{s} + e \mod q$, with $e$ an error from a centered distribution with small standard deviation. SALSA runs in three steps. First, it uses LWE data to train $\mathcal{M}$ to predict $b$ given $\mathbf{a}$. Next SALSA runs a secret recovery algorithm. It feeds $\mathcal{M}$ special values of $\mathbf{a}$, and uses the output $\tilde{b} = \mathcal{M}(\mathbf{a})$ to predict the secret. Finally, SALSA evaluates the guesses $\tilde{\mathbf{s}}$ by verifying that residuals $r = b - \mathbf{a} \cdot \tilde{\mathbf{s}} \mod q$ computed from LWE samples have small standard deviation. If so, $s$ has been recovered and SALSA stops. If not, SALSA returns to step 1 and iterates.

### 4.2    Model Training

SALSA uses LWE instances to train a model that predicts $b$ from $\mathbf{a}$ by minimizing the cross-entropy between the model prediction $b'$ and $b$. The model architecture is a universal transformer [27], in which a shared transformer layer is iterated several times (the output from one iteration is the input to the next). Our base model has two encoder layers, with 1024 dimensions and 32 attention heads, the second layer iterated 2 times, and two decoder layers with 512 dimensions and 8 heads, the second layer iterated 8 times. To limit computation in the shared layer, we use the copy-gate mechanism from [24]. Models are trained using the Adam optimizer with $lr = 10^{-5}$ and 8000 warmup steps.

For inference, we use a beam search with depth 1 (greedy decoding) [40, 65]. At the end of each epoch, we compute model accuracy over a test set of LWE samples. Because of the error added when computing $b = \mathbf{a} \cdot \mathbf{s} + e$, exact prediction of $b$ is not possible. Therefore, we calculate *accuracy within tolerance $\tau$ ($acc_\tau$)*: the proportion of predictions $\tilde{b} = \mathcal{M}(\mathbf{a})$ that fall within $\tau q$ of $b$, i.e. such that $\|b - \tilde{b}\| \leq \tau q$. In practice we set $\tau = 0.1$.

### 4.3    Secret Recovery

We propose two algorithms for recovering $\mathbf{s}$: direct recovery from special values of $\mathbf{a}$, and distinguisher recovery using the binary search to decision reduction (Appendix A.2). For theoretical justification of these, see Appendix C.

**Direct Secret Recovery.** The first technique, based on the LWE search problem, is analogous to a chosen plaintext attack. For each $i \in \mathbb{N}_n$, a guess of the $i$-th coordinate of $\mathbf{s}$ is made by feeding model $\mathcal{M}$ the special value $\mathbf{a_i} = K\mathbf{e_i}$ (all coordinates 0 except the $i$-th), with $K$ a large integer. If $s_i = 0$, and the model $\mathcal{M}$ correctly approximates $b_i = \mathbf{a_i} \cdot \mathbf{s} + e$ from $\mathbf{a_i}$, then we expect $\tilde{b}_i := \mathcal{M}(\mathbf{a}_i)$ to be a small integer; likewise if $s_i = 1$ we expect a large integer. This technique is formalized in Algorithm 1. The $binarize$ function in line 7 is explained in Appendix C. In SALSA, we run direct recovery with 10 different $K$ values in order to yield 10 $\mathbf{s}$ guesses.

---

**Algorithm 1** Direct Secret Recovery

1: **Input:** $\mathcal{M}, K, n$
2: **Output:** secret $s$
3: $p = \mathbf{0}^n$
4: **for** $i = 1, \ldots, n$ **do**
5:    $a = \mathbf{0}^n$; $a_i = K$
6:    $p_i = \mathcal{M}(a)$
7: $s = binarize(p)$
8: **Return:** $s$

---

**Distinguisher Secret Recovery.** The second algorithm for secret recovery is based on the decision-LWE problem. It uses the output of $\mathcal{M}$ to determine if LWE data $(\mathbf{a}, b)$ can be distinguished from randomly generated pairs $(\mathbf{a_r}, b_r)$. The algorithm for distinguisher-based secret recovery is shown in Algorithm 2. At a high level, the algorithm works as follows. Suppose we have $t$ LWE instances $(\mathbf{a}, b)$ and $t$ random instances $(\mathbf{a_r}, b_r)$. For each secret coordinate $s_i$, we transform the $\mathbf{a}$ into $a'_i = a_i + c$, with $c \in \mathbb{Z}_q$ random integers. We then use model $\mathcal{M}$ to compute $\mathcal{M}(\mathbf{a}')$ and $\mathcal{M}(\mathbf{a_r})$. If the model has learned $\mathbf{s}$ and the $i^{th}$ bit of $\mathbf{s}$ is 0, then $\mathcal{M}(\mathbf{a}')$ should be significantly closer to $b$ than $\mathcal{M}(\mathbf{a_r})$ is to $b_r$. Iterating on $i$ allows us to recover the secret bit by bit. SALSA runs the distinguisher recovery algorithm when model $acc_{\tau=0.1}$ is above 30%. This is the theoretical limit for this approach to work.

## 4.4 Secret Verification.

At the end of the recovery step, we have 10 or 11 guesses $\tilde{\mathbf{s}}$ (depending on whether the distinguisher recovery algorithm was run). To verify them, we compute the residuals $r = \mathbf{a} \cdot \tilde{\mathbf{s}} - b \mod q$ for a set of LWE samples $(\mathbf{a}, b)$. If $\mathbf{s}$ is correctly guessed, we have $\tilde{\mathbf{s}} = \mathbf{s}$, so $r = \mathbf{a} \cdot \mathbf{s} - b = e \mod q$ will be distributed as the error $e$ with small standard deviation $\sigma$. If $\tilde{\mathbf{s}} \neq \mathbf{s}$, $r$ will be (approximately) uniformly distributed over $\mathbb{Z}_q$ (because $\mathbf{a} \cdot \tilde{\mathbf{s}}$ and $b$ are uniformly distributed over $\mathbb{Z}_q$), and will have standard deviation $\sigma(r) \approx q/\sqrt{(12)}$. Therefore, we can verify if $\tilde{\mathbf{s}}$ is correct by calculating the standard deviation of the residuals: if it is close to $\sigma$, the standard deviation of error, the secret was recovered. In this paper, $\sigma = 3$ and $q = 251$, so the standard deviation of $r$ will be around 3 if $\tilde{\mathbf{s}} = \mathbf{s}$, and 72.5 if not.

## 5 SALSA Evaluation

In this section, we present our experiments with SALSA. We generate datasets for LWE problems of different sizes, defined by the dimension

---

**Algorithm 2** Distinguisher Secret Recovery

1: **Input:** $\mathcal{M}, n, q, acc_\tau, \tau$
2: **Output:** secret $s$
3: $s = \mathbf{0}^n$
4: $advantage, bound = acc_\tau - 2 \cdot \tau, \tau \cdot q$
5: $t = min\{50, \frac{2}{advantage^2}\}$
6: $\mathbf{A_{LWE}}, \mathbf{B_{LWE}} = LWESamples(t, n, q)$
7: **for** $i = 1, \ldots n$ **do**
8:    $\mathbf{A_{unif}} \sim \mathcal{U}\{0, q-1\}^{n \times t}$
9:    $\mathbf{B_{unif}} \sim \mathcal{U}\{0, q-1\}^t$
10:    $\mathbf{c} \sim \mathcal{U}\{0, q-1\}^t$
11:    $\mathbf{A'_{LWE}} = \mathbf{A_{LWE}}$
12:    $\mathbf{A'_{LWE}}[:, i] = (\mathbf{A_{LWE}}[:, i] + \mathbf{c}) \mod q$
13:    $\widetilde{\mathbf{B_{LWE}}} = \mathcal{M}(\mathbf{A'_{LWE}})$
14:    $\widetilde{\mathbf{B_{unif}}} = \mathcal{M}(\mathbf{A_{unif}})$
15:    $dl = |\widetilde{\mathbf{B_{LWE}}} - \mathbf{B_{LWE}}|$
16:    $du = |\widetilde{\mathbf{B_{unif}}} - \mathbf{B_{unif}}|$
17:    $c_{LWE} = \#\{j \mid dl_j < bound, j \in \mathbb{N}_t\}$
18:    $c_{unif} = \#\{j \mid du_j < bound, j \in \mathbb{N}_t\}$
19:    **if** $(c_{LWE} - c_{unif}) \leq advantage \cdot t/2$ **then**
20:      $s_i = 1$
21: **Return:** $s$

---

and the density of ones in the binary secret. We use gated universal transformers, with two layers in the encoder and decoder. Default dimensions and attention heads in the encoder and decoder are 1024/512 and 16/4, but we vary them as we scale the problems. Models are trained on two NVIDIA Volta 32GB GPUs on an internal cluster.

### 5.1 Data generation

We generate LWE data for SALSA training/evaluation is randomly given the following parameters: dimension $n$, secret density $d$, modulus $q$, encoding base $B$, binary secret $s$, and error distribution $\chi$. For all experiments, we use $q = 251$ and $B = 81$ (see §3.1), fix the error distribution $\chi$ as a discrete Gaussian with $\mu = 0, \sigma = 3$ [4], and generate a random $s$.

We vary the problem size $n$ (the LWE dimension) and the density $d$ (the proportion of ones in the secret) to test attack success and to observe how it scales. For problem size, we experiment with

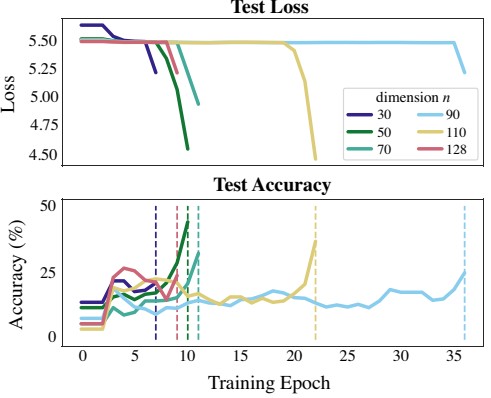

Figure 3: **Full secret recovery**: Curves for loss and $acc_\tau = 0.1$, for varying $n$ with Hamming weight 3. For $n < 100$, model has 1024/512 embedding, 16/4 attention heads. For $n \geq 100$, model has 1536/512 embedding, 32/4 attention heads.

Table 2: **Full secret recovery.** Highest density values at which the secret was recovered for each $n$, $q = 251$. For $n < 100$, the model has 1024/512 embedding, 16/4 attention heads. For $n \geq 100$, the model has 1536/512 embedding, 32/4 attention heads. For comparison, we include the $log_2$ number of possible secrets at each $n/d$ level.

| Dim. $n$ | Density $d$ | $log_2$ samples | $log_2$ secrets | Runtime (hours) |
|---|---|---|---|---|
| 30 | 0.1 | 21.9 | 12.0 | 1.2 |
| | 0.13 | 24.8 | 14.7 | 21 |
| 50 | 0.06 | 22.4 | 14.3 | 5.5 |
| | 0.08 | 25.6 | 17.8 | 18.5 |
| 70 | 0.04 | 22.5 | 15.7 | 4.5 |
| 90 | 0.03 | 24.1 | 16.8 | 35.0 |
| 110 | 0.03 | 21.5 | 17.7 | 32.0 |
| 128 | 0.02 | 22.3 | 18.4 | 23.0 |

$n = 30$ to $n = 128$. For density, we experiment with $0.002 \leq d \leq 0.15$. For a given $n$, we select $d$ so that the Hamming weight of the binary secret ($h = dn$), is larger than 2. Appendix D.2 contains an ablation study of data parameters. We generate data using the RLWE variant of LWE, described in Appendix A. For RLWE problems, each $\mathbf{a}$ is a line of a circulant matrix generated from an initial vector $\in \mathbb{Z}_q^n$. RLWE problems exhibit more structure than LWE due to the use of the circulant matrix, which may help our models learn.

## 5.2 Results

Table 2 presents problem sizes $n$ and densities $d$ for which secrets can be fully recovered, together with the time and the logarithm of number of training samples needed. SALSA can recover binary secrets with Hamming weight 3 for dimensions up to 128. Hamming weight 4 secrets can be recovered for $n < 70$. For context, we include a column "$log_2$ secrets" listing the number of possible secrets for the corresponding Hamming weight. Exhaustive search for the secret may run faster than our current experiments for low Hamming weight. However, an exhaustive search requires knowing the exact hamming weight in advance, or iterating through all possible Hamming weights. Although we have only succeeded unconditionally with low Hamming weight so far, SALSA does not a priori know the secret Hamming weight $h$. It remains to be seen how SALSA will scale to larger $n$ and $h$.

Interestingly, the number of samples required for SALSA to succeed for a fixed Hamming weight is relatively flat as the dimension $n$ increases. However the time needed to recover the secret for a fixed Hamming weight increases with $n$, partly because the length of the input sequence fed into the model is proportional to $n$ even though the number of samples needed remains stable as $n$ grows. This is an important result, because all the data used for training the model must be collected (e.g. via eavesdropping), making sample size an important metric.

For a given $n$, scaling to higher densities requires more time and data, and could not be achieved with the architecture we use for $n > 50$. As $n$ grows, larger models are needed: our standard architecture, with 1024/512 dimensions and 16/4 attention heads (encoder/decoder) was sufficient for $n \leq 90$. For $n > 90$, we needed 1536/512 dimensions and 32/4 attention heads.

Figure 3 illustrates model behavior during training. After an initial burn-in period, the loss curve (top graph) plateaus until the model begins learning the secret. Once loss starts decreasing, model accuracy with $0.1q$ tolerance (bottom graph) increases sharply. Full secret recovery (vertical lines in the bottom graph) happens shortly after, often within one or two epochs. Direct secret recovery accounts for 55% of recoveries, while the distinguisher only accounts for 18% of recoveries (see Appendix C.3). 27% of the time, both methods succeed simultaneously.

One key conclusion from these experiments is that the secret recovery algorithms enable secret recovery long before the transformer has been trained to high accuracy (even before training loss settles at a low level). Frequently, the model only needs *to begin* to learn for the attack to succeed.

Table 3: **Architecture Experiments** We test the effect of model layers, loops, gating, and encoder dimension and report the $log_2$ samples required for secret recovery ($n = 50$, Hamming weight 3).

| Regular vs. UTs (1024/512, 16/4, 8/8) | | Ungated vs. Gated (1024/512, 16/4, 8/8) | | UT Loops (1024/512, 16/4, **X/X**) | | | Encoder Dim. (**X**/512, 16/4, 2/8) | | | Decoder Dim. (1024/**X**, 16/4, 2/8) | | | |
|---|---|---|---|---|---|---|---|---|---|---|---|---|---|
| Regular | UT | Ungated | Gated | 2/8 | 4/4 | 8/2 | 512 | 2048 | 3040 | 256 | 768 | 1024 | 1536 |
| 26.3 | 22.5 | 26.5 | 22.6 | 23.5 | 26.1 | 23.2 | 23.3 | 20.1 | 19.7 | 22.5 | 21.8 | 23.9 | 24.3 |

## 5.3 Experiments with model architecture

SALSA's base model architecture is a Universal Transformer (UT) with a copy-gate mechanism. Table 3 demonstrates the importance of these choices. For problem dimension $n = 50$, replacing the UT by a regular transformer with 8 encoder/decoder layers, or removing the copy-gate mechanism increases the data requirement by a factor of 14. Reducing the number of iterations in the shared layers from 8 to 4 has a similar effect. Reducing the number of iterations in either the encoder or decoder (i.e. from 8/8 to 8/2 or 2/8) may further speed up training. Asymmetric transformers (e.g. large encoder and small decoder) have proved efficient for other math problems, e.g. [37], [17], and asymmetry helps SALSA as well. Table 3 demonstrates that increasing the encoder dimension from 1024 to 3040, while keeping the decoder dimension at 512, results in a 7-fold reduction in sample size. Additional architecture experiments are presented in Appendix D.1.

## 5.4 Increasing dimension and density

To attack real-world LWE problems, SALSA must handle larger dimension $n$ and density $d$. Our experiments with architecture suggest that increasing model size, and especially encoder dimension, is the key factor to scaling $n$. Empirical observations indicate that scaling $d$ is a much harder problem. We hypothesize that this is due to the subset sum modular addition at the core of LWE with binary secrets. For a secret with Hamming weight $h$, the base operation $\mathbf{a} \cdot \mathbf{s} + e \mod q$ is a sum of $h$ integers, followed by a modulus. For small values of $h$, the modulus operation is not always necessary, as the sum might not exceed $q$. As density increases, so does the number of times the sum "wraps around" the modulus, perhaps making larger Hamming weights more difficult to learn. To test this hypothesis, we limited the range of the coordinates in $\mathbf{a}$, so that $a_i < r$, with $r = \alpha q$ and $0.3 < \alpha < 0.7$. For $n = 50$, we recovered secrets with density up to $0.3$, compared to $0.08$ with the full range of coordinates (see Table 4). Density larger than $0.3$ is no longer considered a sparse secret.

## 5.5 Increasing error size

Theoretically for lattice problems to be hard, $\sigma$ should scale with $\sqrt{n}$, although this is often ignored in practice, e.g. [4]. Consequently, we run most SALSA experiments with $\sigma = 3$, a common choice in existing RLWE-based systems. Here, we investigate how SALSA performs as

Table 5: $log_2$ **samples needed for secret recovery when $\sigma = \lfloor \sqrt{n} \rfloor$.** Results averaged over 6 SALSA runs at each $n/\sigma$ level.

| **n/$\sigma$** | 30/5 | 50/7 | 70/8 | 90/9 |
|---|---|---|---|---|
| **logSamples** | 18.0 | 18.5 | 19.3 | 19.6 |

$\sigma$ increases. First, to match the theory, we run experiments where $\sigma = \lfloor \sqrt{n} \rfloor$, $h = 3$ and found that SALSA recovers secrets even as $\sigma$ scales with $n$ (see Table 5, same model architecture as Table 2). Second, we evaluate SALSA's performance for fixed $n/h$ values as $\sigma$ increases. We fix $n = 50$ and $h = 3$ and evaluate for $\sigma$ values up to $\sigma = 24$. Secret recovery succeeds for all tests, although the number of samples required for recovery linearly increases (see Figure 7 in Appendix). For both sets of experiments, we reuse samples up to 10 times.

## 6 SALSA in the Wild

**Problem Size.** Currently, SALSA can recover secrets from LWE samples with $n$ up to 128 and density $d = 0.02$. It can recover higher density secrets for smaller $n$ ($d = 0.08$ when $n = 50$). As mentioned in Section 2.2, sparse binary secrets are used in real world LWE homomorphic encryption, and attacking these implementations is a future goal for SALSA. Admittedly, SALSA must scale to attack larger $n$ before it can break full-strength homomorphic encryption implementations. However, other parameters of full-strength homomorphic encryption such as secret density (the secret vector in HEAAN has $d < 0.002$) and error size ( [4] recommends $\sigma = 3.2$) are within SALSA's reach.

Table 4: **Secret recovery when max a value is bounded.** Results shown are fraction of the secret recovered by SALSA for $n = 50$ with varying $d$ when $a$ values are $\leq p \cdot Q$. Green means that $s$ was fully recovered. Yellow means all of the 1 bits were recovered, but not all 0 bits. Red means SALSA failed. For completeness, we also note the $log_2$ number of possible secrets at each $d$ level.

| $d$ | $log_2$ secrets | Max $a$ value as fraction of $q$ | | | | | | |
|---|---|---|---|---|---|---|---|---|
| | | 0.35 | 0.4 | 0.45 | 0.5 | 0.55 | 0.6 | 0.65 |
| 0.16 | 29.0 | 1.0 | 1.0 | 1.0 | 1.0 | 1.0 | 1.0 | 0.88 |
| 0.18 | 31.2 | 1.0 | 1.0 | 1.0 | 1.0 | 0.82 | 0.86 | 0.84 |
| 0.20 | 33.3 | 1.0 | 1.0 | 1.0 | 1.0 | 1.0 | 0.82 | 0.82 |
| 0.22 | 35.1 | 0.98 | 1.0 | 1.0 | 0.98 | 0.80 | 0.78 | 0.86 |
| 0.24 | 36.8 | 1.0 | 1.0 | 1.0 | 0.98 | 0.78 | 0.78 | 0.80 |
| 0.26 | 38.4 | 1.0 | 1.0 | 0.88 | 0.92 | 0.76 | 0.76 | 0.76 |
| 0.28 | 39.8 | 0.98 | 1.0 | 0.80 | 0.74 | 0.74 | 0.76 | 0.74 |
| 0.30 | 41.0 | 0.98 | 1.0 | 0.93 | 0.76 | 0.72 | 0.74 | 0.74 |

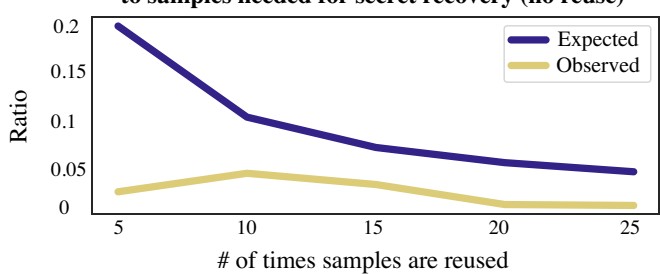

**Ratio of samples needed for secret recovery (with reuse) to samples needed for secret recovery (no reuse)**

Figure 4: **Reusing LWE samples yields a significant decrease in the number of samples needed for secret recovery.** Shown here is the ratio of samples required for secret recovery with reuse to the samples required for secret recovery without reuse, both expected (top curve) and observed (bottom curve, better than expected).

Other LWE-based schemes use secret dimensions that seem achievable given our current results. For example, in the LWE-based public key encryption scheme Crystal-Kyber [9], the secret dimension is $k \times 256$ for $k = \{2, 3, 4\}$, an approachable range for SALSA based on initial results. The LWE-based signature scheme Crystal-Dilithium has similar $n$ sizes [29]. However, these schemes don't use sparse binary secrets, and adapting SALSA nonbinary secrets is an avenue for future work.

**Sample Efficiency.** A key requirement of real-world LWE attacks is sample efficiency. In practice, an attacker will only have access to a small set of LWE instances $(\mathbf{a}, b)$ for a given secret $s$. For instance, in Crystal-Kyber, there are only $(k + 1)n$ LWE instances available with $k = 2$, 3 or 4 and $n = 256$. The experiments in [20, 11] use less than 500 LWE instances. The TU Darmstadt challenge provides $n^2$ LWE instances to attackers.

The $log_2$ $samples$ column of Table 2 lists the number of LWE instances needed for model training. This number is much larger than what is likely available in practice, so it is important to reduce sample requirements. Classical algebraic attacks on LWE require LWE instances to be linearly independent, but *SALSA does not have this limitation.* Thus, we can reduce SALSA's sample use in several ways. First, we can reuse samples during training. Figure 4 confirms that this allows secret recovery with fewer samples. Second, we can use integer linear combinations of given LWE samples to make new samples which have the same secret but a larger error $\sigma$. Appendix E contains the formula for the number of new samples we can generate with this method (up to $2^{42}$ new samples from 100 samples).

**Comparison to Baselines.** Most existing attacks on LWE such as uSVP and dual attack use an algebraic approach that involves building a lattice from LWE instances such that this lattice contains an exceptionally short vector which encodes the secret vector information. Attacking LWE then involves finding the short vector via lattice reduction algorithms like BKZ [22]. For LWE with sparse binary secrets, the main focus of this paper, various techniques can be adapted to make algebraic attacks more efficient. [20, 11] and [23] provide helpful overviews of algebraic attacks on sparse binary secrets. More information about attacks on LWE is in Appendix A.3.

Compared to existing attacks, SALSA's most notable feature is its novelty. We do not claim that to have better runtime, neither do we claim the ability to attack real-world LWE problems (yet). Rather, we introduce a new attack and demonstrate with non-toy successes that transformers can be used to attack LWE. Given our goal, no serious SALSA speedup attempts have been made so far, but a few simple improvements could reduce runtime. First, the slowest step in SALSA is model training, which can be greatly accelerated by distributing it across many GPUs. Second, our transformers are trained from scratch, so pre-training them on such basic tasks as modular arithmetic could save time and data. Finally, the amount of training needed before the secret is recovered depends in large part on the secret guessing algorithms. New algorithms might allow SALSA to recover secrets faster.

Since SALSA does not involve finding the shortest vector in a lattice, it has an advantage over the algebraic attacks – with all LWE parameters fixed and in the range of SALSA, SALSA can attack the LWE problem for a smaller modulus $q$ compared to the algebraic attacks. This is because the target vector is relatively large in the lattice when $q$ is smaller and is harder to find. For instance, in [20], their Table 2 shows that when the block size is $45$, for $n = 90$, their attack does not work for $q$ less than 10 bits, but we can handle $q$ as small as 8 bits (Table 20).

## 7   Related Work

**Use of ML for cryptanalysis.** The fields of cryptanalysis and machine learning are closely related [57]. Both seek to approximate an unknown function $\mathcal{F}$ using data, although the context and techniques for doing so vary significantly between the fields. Because of the similarity between the domains, numerous proposals have tried to leverage ML for cryptanalysis. ML-based attacks have been proposed against a number of cryptographic schemes, including block ciphers [3, 63, 38, 10, 30, 12, 21], hash functions [31], and substitution ciphers [1, 64, 8]. Although our work is the first to use recurrent neural networks for lattice cryptanalysis, prior work has used them for other cryptographic tasks. For example, [32] showed that LSTMs can learn the decryption function for polyalphabetic ciphers like Enigma. Follow-up works used variants of LSTMs, including transformers, to successfully attack other substitution ciphers [1, 64, 8].

**Use of transformers for mathematics.** The use of language models to solve problems of mathematics has received much attention in recent years. A first line of research explores math problems set up in natural language. [59] investigated their relative difficulty, using LSTM [34] and transformers, while [33] showed large transformers could achieve high accuracy on elementary/high school problems. A second line explores various applications of transformers on formalized symbolic problems. [42] showed that symbolic math problem could be solved to state-of-the-art accuracy with transformers. [68] discussed their limits when generalizing out of their training distribution. Transformers have been applied to dynamical systems [18], transport graphs [19], theorem proving [54], SAT solving [61], and symbolic regression [13, 26]. A third line of research focuses on arithmetic/numerical computations and has had slower progress. [52] and [51] discussed the difficulty of performing arithmetic operations with language models. Bespoke network architectures have been proposed for arithmetic operations [35, 66], and transformers were used for addition and similar operations [55]. [17] showed that transformers can learn numerical computations, such as linear algebra, and introduced the shallow models with shared layers used in this paper.

**Learnability of LWE.** Shalev-Shwartz, Shamir and Shammah [60] showed that some problems of modular arithmetic cannot be solved by gradient descent. Because it uses a transformer to learn the secret, by predicting $b$ from $a$, SALSA's scalability hinges on the solvability of LWE by gradient descent. Here, we provide our perspective on this question.

The LWE problem amounts to (discrete) linear regression on an $n$-dimensional torus with radius $q$. When $q$ is infinite, LWE is pure linear regression, and gradient methods will succeed. For finite $q$, gradients are informative unless the model's prediction lies on the opposite side of the torus from the true solution (e.g. $q = 100$, true value is $0$, prediction is $50$). At this point, the projection of the gradient along the coordinate axis is uninformative, since either direction on the torus points towards the true solution. For large $q$, this situation is uncommon, so gradient methods should work. For very small $q$, this will happen more often, and gradient methods will be perturbed. In the degenerate case $q = 2$, featured in [60], gradient methods will always fail.

For the value $q = 251$ used in SALSA, gradient methods can recover secrets (i.e. solve the modular linear regression problem) for $n$ up to $128$ so far. Except for the case of very small $q$, it is unlikely

that LWE belongs to the "failing gradient" class of problems described in [60]. Any learnability problems should disappear altogether as $q$ increases.

Interestingly, this intuition about the size of $q$ also appears in the classical lattice reduction approach to solving LWE. Laine and Lauter [41] use LLL to demonstrate concrete polynomial-time attacks against lattice problems for large $q$. They also explore the boundary of where these attacks start to fail for smaller $q$. The intuition is that when $q$ is large, LLL directly finds a vector which is "small enough" compared to $q$ to break the system. When $q$ is small, LLL does not find a short enough vector. This is further explored in [20], which gives concrete running times for generalizations of LLL and quantifies the size of $q$ where these attacks start to fail.

## 8   Conclusion

**Discussion.**  In this paper, we demonstrate that transformers can be trained to perform modular arithmetic. Building on this capability, we design SALSA, a method for attacking the LWE problem with binary secrets, a hardness assumption at the foundation of many lattice-based cryptosystems. We show that SALSA can break LWE problems of medium dimension (up to $n = 128$), which is in the same range as problems in the Darmstadt challenge [15], although we restrict to the easier case of sparse binary secrets. This is the first paper to use transformers to solve hard problems in lattice-based cryptography. Future work will attempt to scale up SALSA to attack higher dimensional lattices with more general secret distributions.

The key to scaling up to larger lattice dimensions seems to be to increase the model size, especially the dimensions, the number of attention heads, and possibly the depth. Large architectures should scale to higher dimensional lattices such as $n = 256$ which is used in practice. Density, on the other hand, is constrained by the performance of transformers on modular arithmetic. Better representations of finite fields could improve transformer performance on these tasks. Finally, our secret guessing algorithms enable SALSA to recover secrets from low-accuracy transformers, therefore reducing the data and time needed for the attack. Extending these algorithms to take advantage of partial learning should result in better performance.

**Ethics and Broader Impact.**  The primary value of this work is in alerting the cryptographic and ML communities to the risk of ML-based attacks on PQC. Even if current attacks do not succeed, we believe that providing early warning of potential threats is critical. However, we emphasize that SALSA represents a proof of concept that cannot be used against real-world implementations (i.e. the PQC schemes which NIST standardized on July 5, 2022). Additional scaling work would be necessary before these techniques would be relevant to attacking real-world cryptosystems.

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
