# Supplementary Materials

## A    Further Details of LWE

### A.1    Ring Learning with Errors (§2)

We now define RLWE samples and explain how to get LWE instances from them. Let $n$ be a power of 2, and let $R_q = \mathbb{Z}_q[x]/(x^n + 1)$ be the set of polynomials whose degrees are at most $n - 1$ and coefficients are from $\mathbb{Z}_q$. The set $R_q$ forms a ring with additions and multiplications defined as the usual polynomial additions and multiplications in $\mathbb{Z}_p[x]$ modulo $x^n + 1$. One RLWE sample refers to the pair

$$(a(x), b(x) := a(x) \cdot s(x) + e(x)),$$

where $s(x) \in R_q$ is the secret and $e(x) \in R_q$ is the error with coefficients subject to the error distribution.

Let $\mathbf{a}, \mathbf{s}$ and $\mathbf{e} \in \mathbb{Z}_q^n$ be the coefficient vectors of $a(x)$, $s(x)$ and $e(x)$. Then the coefficient vector $\mathbf{b}$ of $b(x)$ can be obtained via the formula

$$\mathbf{b} = \mathbf{A}_{a(x)}^{\text{circ}} \cdot \mathbf{s} + \mathbf{e},$$

here $\mathbf{A}_{a(x)}^{\text{circ}}$ represents the $n \times n$ generalized circulant matrix of $a(x)$. Precisely, let $a(x) = a_0 + a_1 x + \ldots + a_{n-2}x^{n-2} + a_{n-1}x^{n-1}$, then $\mathbf{a} = (a_0, a_1, \ldots, a_{n-2}, a_{n-1})$ and

$$\mathbf{A}_{a(x)}^{\text{circ}} = \begin{bmatrix} a_0 & -a_{n-1} & -a_{n-2} & \ldots & -a_1 \\ a_1 & a_0 & -a_{n-1} & \ldots & -a_2 \\ a_2 & a_1 & a_0 & \ldots & -a_3 \\ \vdots & \vdots & \vdots & \ddots & \vdots \\ a_{n-1} & a_{n-2} & a_{n-3} & \ldots & a_0 \end{bmatrix}.$$

Therefore, one RLWE sample gives rise to $n$ LWE instances by taking the rows of $\mathbf{A}_{a(x)}^{\text{circ}}$ and the corresponding entries in $\mathbf{b}$.

### A.2    Search to Decision Reduction for Binary Secrets (§2)

We give a proof of the search binary-LWE to decisional binary-LWE reduction. This is a simple adaption of the reduction in [56] to the binary secrets case. We call an algorithm a $(T, \gamma)$-distinguisher for two probability distributions $\mathcal{D}_0, \mathcal{D}_1$ if it runs in time $T$ and has a distinguishing advantage $\gamma$. We use $\mathbf{LWE}_{n,m,q,\chi}$ to denote the LWE problem which has secret dimension $n$, $m$ LWE instances, modulus $q$ and the secret distribution $\chi$.

**Theorem A.1.**  *If there is a $(T, \gamma)$-distinguisher for decisional binary-$\mathbf{LWE}_{n,m,q,\chi}$, then there is a $T' = \widetilde{O}(Tn/\gamma^2)$-time algorithm that solves search binary-$\mathbf{LWE}_{n,m',q,\chi}$ with probability $1 - o(1)$, where $m' = \widetilde{O}(m/\gamma^2)$.*

*Proof.*  Let $\mathbf{s} = (s_1, \ldots, s_n)$ with $s_i \in \{0, 1\}$. We demonstrate the strategy of recovering $s_1$, and the rest of the secret coordinates can be recovered in the same way. Let $m' = \widetilde{O}(1/\gamma^2)m$, given an LWE sample $(\mathbf{A}, \mathbf{b})$ where $\mathbf{A} \in \mathbb{Z}_q^{m' \times n}, \mathbf{b} \in \mathbb{Z}_q^{m'}$, we compute a pair $(\mathbf{A}', \mathbf{b}')$ as follows:

$$\mathbf{A}' = \mathbf{A} +' \mathbf{c}, \quad \mathbf{b}' = \mathbf{b}.$$

Here $\mathbf{c} \in \mathbb{Z}_q^{m'}$ is sampled uniformly and the symbol "$+'$" means that we are adding $\mathbf{c}$ to the first column of $\mathbf{A}$. One verifies by the definition of LWE that if $s_1 = 0$, then the pair $(\mathbf{A}', \mathbf{b}')$ would be LWE samples with the same error distribution. Otherwise, the pair $(\mathbf{A}', \mathbf{b}')$ would be uniformly random in $\mathbb{Z}_q^{m' \times n} \times \mathbb{Z}_q^{m'}$. We then feed the pair $(\mathbf{A}', \mathbf{b}')$ to the $(T, \gamma)$-distinguisher for $\mathbf{LWE}_{n,m,q,\chi}$, and we need to running the distinguisher $m'/m = \widetilde{O}(1/\gamma^2)$ times given the number of instances. Since the advantage of this distinguisher is $\gamma$ with $m$ LWE instances, and we are feeding it $m'$ LWE instances, it follows from the Chernoff bound that if the majority of the outputs are "LWE", then

the pair $(\mathbf{A}', \mathbf{b}')$ is an LWE sample and therefore $s_1 = 0$. If not, $s_1 = 1$. Guessing one coordinate requires running the distinguisher $\widetilde{O}(1/\gamma^2)$ times, therefore, this search to reduction algorithm takes time $T' = \widetilde{O}(Tn/\gamma^2)$. Note that we can use the same $m'$ LWE instances for each coordinate, therefore it requires $m' = \widetilde{O}(m/\gamma^2)$ samples to recover all the secret coordinates. $\qquad\square$

### A.3 Overview of Attacks on LWE

Typically, attacks on the LWE problem use an algebraic approach and involve lattice reduction algorithms such as BKZ [22]. The LWE problem can be turned into a BDD problem (Bounded Distance Decoding) by considering the lattice generated by LWE instances, and BDD can be solved by Babai's Nearest Plane algorithm [44] or the pruned enumeration [45], this is known as the primal BDD attack. The primal uSVP attack constructs a lattice via Kannan's embedding technique [36] whose unique shortest vector encodes the secret information. The Dual attack [48] finds a short vector in the dual lattice which can be used to distinguish the LWE samples from random samples. Moreover, there are also attacks that do not use lattice reduction. For instance, the BKW style attack [5] uses combinatorial methods; however, this assumes access to an unbounded number of LWE samples.

Binary and ternary secret distributions are widely used in homomorphic encryption schemes. In fact, many implementations even use a sparse secret with Hamming weight $h$. In [14] and [47], both papers give reductions of binary-LWE to hard lattice problems, implying the hardness of binary-LWE. Specifically, the $(n, q)$-binary-LWE problem is related to a $(n/t, q)$-LWE problem where $t = O(\log(q))$. For example, if $n = 256$ is a hard case for uniform secret, we can be confident that binary-LWE is hard for $n = 256 \log(256) = 2048$. But [11] refines this analysis and gives an attack against binary-LWE. Their experimental results suggest that increasing the secret dimension by a $\log(\log(n))$ factor might be already enough to achieve the same security level for the corresponding LWE problem with uniform secrets.

Let us now turn to the attacks on (sparse) binary/ternary secrets. The uSVP attack is adapted to binary/ternary secrets in [11], where a balanced version of Kannan's embedding is considered. This new embedding increases the volume of the lattice and hence the chance that lattice reduction algorithms will return the shortest vector. The Dual attack for small secret is considered in [6] where the BKW-style techniques are combined. The BKW algorithm itself also has a binary/ternary-LWE variant [7]. Moreover, several additional attacks are known which can exploit the sparsity of an LWE secret, such as [16, 23] . All of these techniques use a combinatorial search in some dimension $d$, and then follow by solving a lattice problem in dimension $n - d$. For sparse secrets, this is usually more efficient than solving the original lattice problem in dimension $n$.

## B  Additional Modular Arithmetic Results (§3)

Table 6: $q$ values used in our experiments

| $\lceil \log_2(q) \rceil$ | $q$ | $\lceil \log_2(q) \rceil$ | $q$ |
|---|---|---|---|
| 5 | 19, 29 | 18 | 147647, 222553 |
| 6 | 37, 59 | 19 | 397921, 305423 |
| 7 | 67, 113 | 20 | 842779, 682289 |
| 8 | 251, 173 | 21 | 1489513, 1152667 |
| 9 | 367, 443 | 22 | 3578353, 2772311 |
| 10 | 967, 683 | 23 | 6139999, 5140357 |
| 11 | 1471, 1949 | 24 | 13609319, 14376667 |
| 12 | 3217, 2221 | 25 | 31992319, 28766623 |
| 13 | 6421, 4297 | 26 | 41223389, 38589427 |
| 14 | 11197, 12197 | 27 | 94056013, 115406527 |
| 15 | 20663, 24659 | 28 | 179067461, 155321527 |
| 16 | 42899, 54647 | 29 | 274887787, 504470789 |
| 17 | 130769, 115301 | 30 | 642234707, 845813581 |

Here, we provide additional information on our single and multidimensional modular arithmetic experiments from §3.1. Before presenting experimental results, we first highlight two useful tables. Table 6 shows the $q$ values used in our integer and multi-dimension modular arithmetic problems. Table 7 is an expanded version of Table 1 in the main paper body. It shows how the $\log_2$ samples required for success changes with the base representation for the input/output, but includes additional values of base $B$ (secret is fixed at $728$).

Table 7: Base-2 logarithm of the number of examples needed to reach $95\%$ accuracy, for different values of $\lceil \log_2(q) \rceil$ and bases.

| $\lceil \mathbf{log_2}(q) \rceil$ | **Base** | | | | | | | | | | | | |
|---|---|---|---|---|---|---|---|---|---|---|---|---|---|
| | 2 | 3 | 4 | 5 | 7 | 17 | 24 | 27 | 30 | 31 | 63 | **81** | 128 |
| 15 | 23 | 21 | 21 | 23 | 22 | 20 | 20 | 23 | 22 | 21 | 21 | **20** | 20 |
| 16 | 24 | 22 | 23 | 22 | 22 | 23 | 22 | 22 | 22 | 23 | 22 | **22** | 21 |
| 17 | - | 23 | 24 | 25 | 22 | 26 | 23 | 24 | 22 | 24 | 23 | **22** | 22 |
| 18 | - | 23 | 23 | 25 | 23 | - | 23 | 24 | 25 | - | 23 | **22** | 22 |
| 19 | - | 23 | - | - | 25 | 23 | 25 | 24 | - | - | 25 | **25** | 24 |
| 20 | - | - | - | - | - | 24 | 25 | 24 | 26 | - | - | **24** | 25 |
| 21 | - | 24 | - | - | 25 | - | - | - | - | - | - | - | 25 |
| 22 | - | - | - | - | - | - | - | 25 | - | 26 | - | - | 25 |
| 23 | - | - | - | - | - | - | - | - | - | - | 25 | - | - |
| 24 | - | - | - | - | - | - | - | - | - | - | - | - | - |

**Base vs. Secret.** We empirically observe that the base $B$ used for integer representation in our experiments may provide side-channel information about the secret $s$ in the 1D case. For example, in Table 8, when the secret value is 729, bases 3, 9, 27, 729 and 3332 all enable solutions with much higher $q$ (8 times higher than the next highest result). Nearly all these are powers of $3^3$ as is the secret $729 = 3^6$. In the table, one can see that these same bases provide similar (though not as significant) "boosts" in $q$ for secrets on either side of 729 (e.g. 728, 730), as well as for $720 = 3^6 - 3^2$. Based on these results, we speculate that when training on $(a, b)$ pairs with an unknown secret $s$, testing on different bases and observing model performance may allow some insight into $s$'s prime factors. More theoretical and empirical work is needed to verify this connection.

**Ablation over transformer parameter choices.** We provide additional experiments on model architecture, specifically examining the effect of model layers, optimizer, embedding dimension and batch size on integer modular inversion performance. Tables 9-12 show ablation studies for the 1D modular arithmetic task, where entries are of the form (best $\log_2(q)$/$\log_2(samples)$), e.g. the highest modulus achieved and the number of training samples needed to achieve this. The best results, meaning the highest $q$ with the lowest $\log_2(samples)$, are in **bold**. For all experiments, we use the base architecture of 2 encoder/decoder layers, 512 encoder/decoder embedding dimension, and 8/8 attention heads (as in Section 3.1) and note what architecture element changes in the table heading.

We find that shallower transformers (e.g 2 layers, see Table 9) work best, allowing problem solutions with a much higher $q$ especially when the base $B$ is large. The AdamCosine optimizer (Table 10) generally worked best, but required a smaller batch size for success with larger base. For smaller bases, a smaller embedding dimension of 128 performed better (Table 11), but increasing base size and dimension simultaneously yielded good performance. Results on batch size (Table 12) do not show a strong trend.

## C    Additional information on SALSA Secret Recovery (§4.3)

Here, we provide additional information about SALSA's two secret recovery algorithms.

---

[3] And 3332 can easily be written out as a sum of powers of 3, e.g. $3332 = 3^8 - 3^7 - 3^6 - 3^5 - 3^4 + 3^2 + 3 - 1$.

Table 8: Relationship between base and secret. Numbers in table represent the highest $\log_2(q)$ value achieved for a particular base/secret combo. Values of $\log_2(q) >= 23$, indicating high performance, are **bold**.

| Base | Secret value | | | | | | | | | | |
|---|---|---|---|---|---|---|---|---|---|---|---|
| | 720 | 721 | 722 | 723 | 724 | 725 | 726 | 727 | 728 | 729 | 730 |
| 2 | - | 18 | 16 | - | - | - | 16 | 16 | - | 16 | - |
| 3 | 19 | 16 | 18 | 16 | 18 | 18 | 19 | 18 | 21 | **24** | 20 |
| 4 | 18 | 18 | 18 | 18 | 18 | - | - | - | 18 | | 18 |
| 5 | 18 | 17 | 16 | 16 | 16 | 18 | 18 | 17 | 16 | 19 | 16 |
| 7 | - | 18 | - | 18 | - | 20 | - | - | 19 | 18 | - |
| 9 | **23** | 18 | 18 | 18 | 18 | 18 | 18 | 21 | 18 | **24** | **23** |
| 11 | 20 | 20 | - | 19 | 21 | 21 | 21 | 20 | - | - | 19 |
| 17 | 18 | 18 | - | 19 | 19 | 18 | 18 | | 20 | 20 | - |
| 27 | **23** | - | **23** | 18 | 18 | 18 | 21 | 22 | 21 | **24** | 22 |
| 28 | - | 20 | 18 | - | 20 | 18 | - | 19 | **23** | 18 | - |
| 49 | 18 | 22 | 18 | 19 | 21 | 18 | - | 18 | 19 | 18 | 18 |
| 63 | 20 | 21 | 18 | 20 | 19 | 19 | 18 | 19 | 18 | - | - |
| 128 | 20 | - | 18 | 22 | 20 | - | 19 | 18 | 19 | 19 | 19 |
| 729 | 18 | 20 | 19 | 18 | 19 | 21 | 19 | 19 | 18 | **25** | 18 |
| 3332 | 22 | 22 | 22 | **23** | 22 | 22 | 21 | 22 | **23** | **23** | 22 |

Table 9: 1D case: Ablation over number of transformer layers.

| Base | # Transformer Layers | | |
|---|---|---|---|
| | 2 | 4 | 6 |
| 27 | 19/24 | 18/27 | 20/25 |
| 63 | 18/25 | 16/25 | 15/22 |
| 3332 | **23/26** | 23/- | 18/22 |

Table 10: 1D case: Ablation over optimizers. Parenthetical denotes (# warmup steps, learning rate); * = batch size 128.

| Base | Optimizer | | |
|---|---|---|---|
| | Adam $(0, 5e^{-5})$ | Adam $(3000, 5e^{-5})$ | AdamCosine $(3000, 1e^{-5})$ |
| 27 | 18/26 | 19/24 | **22/27** |
| 3332 | **23/26** | 23/27 | 22/26 |
| 3332* | 23/26 | 23/26 | **23/25** |

Table 11: 1D case: Ablation over embedding.

| Base | Embedding Dimension | | | |
|---|---|---|---|---|
| | 512 | 256 | 128 | 64 |
| 3 | 21/25 | 21/24 | **22/26** | 19/- |
| 27 | 23/26 | 23/26 | **23/25** | 19/26 |
| 63 | **23/27** | 18/24 | 19/27 | 18/26 |
| 3332 | **23/25** | 23/26 | 23/26 | 23/27 |

Table 12: 1D case: Ablation over training batch size.

| Base | Batch size | | | | |
|---|---|---|---|---|---|
| | 64 | 96 | 128 | 192 | 256 |
| 3 | 21/26 | 21/25 | 21/26 | 22/26 | **23/26** |
| 27 | 21/25 | **24/27** | 22/27 | 23/26 | 24/28 |
| 63 | - | 20/27 | **23/25** | - | 23/26 |
| 3332 | 23/26 | 23/26 | **23/25** | **23/25** | 23/26 |

## C.1 Direct Secret Recovery

**Recovering Secrets from Predictions.** During the direct secret recovery phase, we must transform model predictions from sequences on integers in base $B$ into binary secrets. This transformation is denoted by the $binarize$ function on line 7 of Algorithm 1. We first decode the $n$ (one for each special **a** input) predictions into integers and concatenate the predictions into one $n$-long vector $\tilde{s}$. Then, we use each of the following methods to binarize this vector: mean comparison, softmax mean comparison, and mode comparison.

- The mean comparison method takes the mean of $\tilde{s}$ and computes *two* potential secrets from it via the following function: $f_{01}(\tilde{s})$ sets all elements above the mean to be 0 and below it to be 1, and $f_{10}(\tilde{s})$ sets all elements above the mean to be 1 and below it to be 0.
- To use the softmax mean comparison, we first take the softmax of $\tilde{s}$. We then take the mean of $\tilde{s}$ and use the same binarization method as before to get two secret predictions.

- The mode comparison method is similar but instead of taking the mean, it uses the mode of $\tilde{\mathbf{s}}$ as the divider between 0 and 1 methods.

Altogether, these binarization methods produce six secret guesses. In our SALSA evaluation, all of these are compared against the true secret $\mathbf{s}$, and the number of matching bits is reported. If $\tilde{\mathbf{s}}$ fully matches $\mathbf{s}$, model training is stopped. When $\mathbf{s}$ is not available for comparison, the methods in §4.4 can be used to verify $\tilde{\mathbf{s}}$'s correctness.

**K values.** At the end of each epoch, we use $10$ $K$ values for direct secret guessing, $5$ of which are fixed and $5$ of which are randomly generated. The fixed $K$ values are $K = [239145, 42899, q - 1, 3q + 7, 42900]$, while the random $K$ values are chosen from the range $(q, 10q)$.

## C.2 Distinguisher-Based Secret Recovery

Here, we provide more details on the parameters and subroutines used in Algorithm 2.

- $\tau$: This parameter sets the bound on $q$ that will be used for the distinguisher computation. In our experiments, we set $\tau = 0.1$.
- $acc_\tau$: This denotes the distinguisher advantage. Let $acc_\tau$ denote the proportion of model predictions which fall within the chosen tolerance (e.g. accuracy within tolerance as described in §4.2), then $advantage = acc_\tau - 2 * \tau$.
- $LWESamples(t, n, q)$ is a subroutine that returns LWE samples $(\mathbf{A}, \mathbf{b}) \in \mathbb{Z}_q^{n \times t} \times \mathbb{Z}_q^t$, note that now columns of $\mathbf{A}$ corresponds to LWE instances.

## C.3 Secret Recovery in Practice

Empirically, we observe that direct secret recovery recovers the secret more quickly than the distinguisher-based method. Figure 5 plots counts of which technique provided the successful secret guess for 120 SALSA runs with varying $n$ and $d$. The direct secret guessing method success $> 90\%$ of the time, but occasionally the distinguisher is able to get the secret first. Occasionally, both methods simultaneously recover the secret.

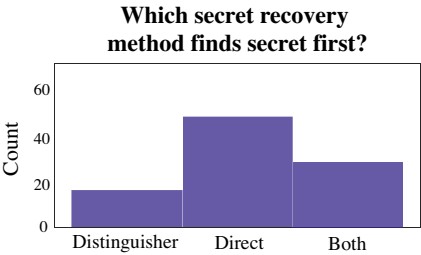

Figure 5: Frequency counts of which secret recovery method succeeds first for 90 successful SALSA runs.

# D   Additional SALSA Results (§5.2)

## D.1   Effect of Architecture for (R)LWE Attacks

Several key architecture choices determine SALSA's ability to recover secrets with higher $n$ and $d$, namely the encoder and decoder dimension as well as the number of attention heads. Other architecture choices determine the time to solution but not the complexity of problems SALSA could solve. For example, universal transformers (UT) are more sample efficient than regular transformers. Using gated loops in the UT with more loops on the decoder than the encoder reduced both model training time and the number of samples needed. Here, we present ablation results for all these architectural choices.

**Universal Transformers vs. Regular Transformers.** First, we see if universal transformers improve experimental efficiency or success. We run dueling experiments on medium size problems ($N = 50$, $d = 0.06$, $q = 251$, $B_{in}/B_{out} = 81$). For one experiment, we use regular transformers with between

Table 13: **Transformers vs UTs.** Ratio of training samples required for success for UTs with X/X encoder/decoder loops vs. regular transformers with X/X encoder/decoder layers.

| Encoder Loops/Layers | Decoder Loops/Layers | | |
| | 2 | 4 | 8 |
|---|---|---|---|
| 2 | 1.2 | 4.7 | 0.8 |
| 4 | 0.7 | 0.4 | 0.6 |
| 8 | 0.1 | 0.1 | 0.1 |

Table 14: **Gated vs Ungated UTs.** Ratio of training samples required for success for gated UTs with X/X encoder/decoder loops vs. ungated UTs with same loop numbers.

| Encoder Loops | Decoder Loops | | |
| | 2 | 4 | 8 |
|---|---|---|---|
| 2 | 1.0 | 1.3 | 0.3 |
| 4 | 0.3 | 0.3 | 0.3 |
| 8 | 0.1 | 0.1 | 0.1 |

Table 15: **Loops.** Average $\log_2$ of training samples required for $N = 50$, $h = 3$, $q = 251$, $base_{in}/base_{out} = 81$ as loops vary.

| Encoder Loops | Decoder Loops | | |
| | 2 | 4 | 8 |
|---|---|---|---|
| 2 | 23.5 | 25.4 | 23.4 |
| 4 | 23.3 | 24.2 | 24.4 |
| 8 | 23.1 | 22.3 | 22.5 |

2 and 8 encoder/decoder layers. For the other, we use gated universal transformers with between 2 and 8 loops. We then compare the relative number of training samples needed to achieve success for both methods. As Table 13 shows, when model size increases, universal transformers prove more sample efficient, so we use them exclusively.

**Gated vs Ungated UTs.** To understand the effect of gating on sample efficiency, we run two experiments with medium-size problems ($N = 50$, $d = 0.06$, $q = 251$, $B_{in}/B_{out} = 81$). For both experiments, we use universal transformers with between 2 and 8 loops on the encoder/decoder. We use gates for one experiment but not for the other. As Table 14 shows, gated UTs are much more sample-efficient.

**Number of Loops.** Table 15 shows the average number of samples needed to recover the secret for $N = 50$, $h = 3$ with different numbers of encoder/decoder loops (dimension=1025/512, heads=16/4). There is a tradeoff between adding loops and increasing computation time, particularly for encoder loops as $n$ increases. In our experiments, we elect to use 2 encoder loops and 8 decoder loops due to the significant training time needed for high $n$ values with more encoder loops.

**Encoder/Decoder Dimension.** Table 3 in §5.2 demonstrates how increasing the encoder size and decreasing the decoder sizes enables secret recovery with fewer samples. Here, we explore the effect of encoder and decoder dimension on the $n$ and hamming weight of secrets SALSA can recover. Our results, shown in Tables 16 and 17, follow the same pattern as before: higher encoder dimension and lower decoder dimension allow us to recover secrets with higher $n$. Furthermore, for $n = 30$, higher encoder dimension and/or smaller decoder dimension allows recovery of hamming weight 4 secrets.

Table 16: **Ablation over Encoder Dimension.** Proportion of secret bits recovered for varying $n$ and encoder dimension. For all experiments, we fix decoder dimension to be 512, 2/2 layers, 2/8 loops. Green means secret was guessed, yellow means all 1s, but not all 0s, were guessed, and red means SALSA failed.

| $n$ | Encoder Dimension (hamming=3) | | | | Encoder Dimension (hamming=4) | | | |
| | 512 | 1024 | 2048 | 3040 | 512 | 1024 | 2048 | 3040 |
|---|---|---|---|---|---|---|---|---|
| 30 | 1.0 | 1.0 | 1.0 | 1.0 | 0.87 | 1.0 | 1.0 | 1.0 |
| 50 | 1.0 | 1.0 | 1.0 | 1.0 | 0.94 | 0.94 | 0.94 | 0.94 |
| 70 | 0.97 | 1.0 | 1.0 | 1.0 | 0.96 | 0.97 | 0.94 | 0.96 |
| 90 | 0.97 | 0.98 | 1.0 | 1.0 | 0.96 | 0.96 | 0.97 | 0.97 |

**Attention Heads.** We run experiments for varying $n$ with 2 encoder/decoder layers, 1024/512 embedding dimension, 2/8 encoder/decoder loops, and varying attention heads to observe the impact of attention heads on SALSA's success. Increasing the number of encoder attention heads while keeping decoder heads at 4 allows SALSA to recover secrets for $n > 70$ (Table 18), although it slightly increases the number of samples needed for recovery (Table 19). Increasing the number of decoder heads increases the number of samples needed but does not provide the same scale-up for $n$.

Table 17: **Ablation over Decoder Dimension.** Proportion of secret bits recovered for varying $n$ and encoder dimension. For all experiments, we fix encoder dimension to be 1024, 2/2 layers, 2/8 loops. Green means secret was guessed, yellow means all 1s, but not all 0s, were guessed, and red means SALSA failed.

| $n$ | Decoder Dimension (hamming=3) | | | | Decoder Dimension (hamming=4) | | | |
|---|---|---|---|---|---|---|---|---|
| | 256 | 768 | 1024 | 1536 | 256 | 768 | 1024 | 1536 |
| 30 | 1.0 | 1.0 | 1.0 | 1.0 | 1.0 | 1.0 | 0.90 | 0.87 |
| 50 | 1.0 | 1.0 | 1.0 | 0.94 | 0.94 | 0.92 | 0.92 | 0.92 |
| 70 | 1.0 | 1.0 | 1.0 | 0.96 | 0.96 | 0.94 | 0.94 | 0.94 |
| 90 | 1.0 | 0.97 | 0.97 | 0.97 | 0.97 | 0.96 | 0.97 | - |

Table 18: **Attention Heads: Effect on secret recovery.** Table of success for varying $n$ with hamming 3 for encoder/decoder head combinations. Green means secret was guessed, yellow means all 1s, but not all 0s, were guessed.

| N | Encoder/Decoder Heads | | | | | | | |
|---|---|---|---|---|---|---|---|---|
| | 8/8 | 16/4 | 16/8 | 16/16 | 32/4 | 32/8 | 32/16 | 32/32 |
| 30 | 1.0 | 1.0 | 1.0 | 1.0 | 1.0 | 1.0 | 1.0 | 1.0 |
| 50 | 1.0 | 1.0 | 1.0 | 1.0 | 1.0 | 1.0 | 1.0 | 1.0 |
| 70 | 1.0 | 1.0 | 1.0 | 1.0 | 1.0 | 1.0 | 1.0 | 1.0 |
| 90 | 0.97 | 1.0 | 0.97 | 0.99 | 1.0 | 0.98 | 0.98 | 0.97 |

### D.2 Effect of data parameters for (R)LWE Attacks

Complex relationships between $N$, $q$, $B$, and $d$ affect SALSA's ability to fully recover secrets. Here, we explore these relationships, with success measured by the proportion of secret bits recovered. Table 20 shows SALSA's performance as $n$ and $q$ vary with fixed hamming weight 3. SALSA performs better for smaller and larger values of $q$, but struggles on mid-size ones across all $N$ values (when hamming weight is held constant). This stands in contrast to traditional attacks on LWE, which only work well when $q$ is large [20]. Table 21 shows the interactions between $q$ and $d$ with fixed $n = 50$. Here, we find that varying $q$ does not increase the density of secrets recovered by SALSA. Finally, Table 22 shows the $\log_2$ samples needed for secret recovery with different input/output bases with $n = 50$ and hamming weight 3. The secret is recovered for all input/output base pairs except for $B_{in} = 17$, $B_{out} = 3$, and using a higher input base reduces the $\log_2$ samples needed for recovery.

### D.3 Effect of training parameters on (R)LWE Attacks

We experiment with numerous training parameters to optimize SALSA's performance (e.g. optimizer, learning rate, floating point precision). While most of these settings do not substantively change SALSA's overall performance, we find that batch size has a significant impact on SALSA's sample efficiency. In experiments with $n = 50$ and Hamming weight 3, small batch sizes, e.g. $< 50$, allow recovery of secrets with much fewer samples, as shown in Figure 6. The same model architecture is used as for $n = 50$ in Table 2.

## E Improving SALSA's Sample Efficiency

Here, we provide both theoretical and empirical results highlighting ways to improve SALSA's sample efficiency.

**Generating New Samples.** We explain how to generate new LWE samples from existing ones via linear combinations. Assume we have access to $m$ LWE samples. Suppose a SALSA model can still learn from samples following a family of Gaussian distributions with standard deviations less than $N\sigma$, where $\sigma$ is the standard deviation of the original LWE error distribution. The number of new samples we could make is equal to the number of vectors $\mathbf{v} = (v_1, \ldots, v_m)^T \in \mathbb{Z}^m$ such that

Table 19: **Attention Heads: Effect on** $\log_2$ **samples.** We test the effect of attention heads and report the $\log_2$ samples required to recover the secret in each setting. Experiments are run with $n = 50$, hamming 3.

| Attention Heads (1024/512, X/X, 2/8) | | |
|---|---|---|
| 8/8 | 16/4,8,16 | 32/4,8,16,32 |
| 22.4 | 22.8, 22.9, 23.2 | 23.0, 23.1, 23.7, 24.7 |

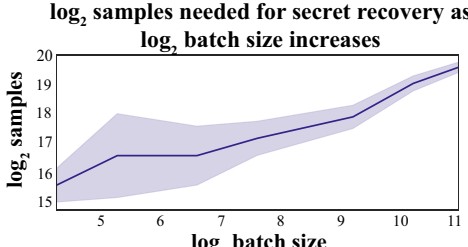

Figure 6: **Batch size and sample efficiency.** Smaller batch sizes allow SALSA to recover secrets faster. Experiments are run with $n = 50$, Hamming weight 3.

$\sum_{i=1}^{m} |v_i| \le N^2$. For simplicity, assume that $v_i's$ are nonnegative. Then, there are $\sum_{n=1}^{N^2} \frac{(m+n-1)!}{(m-1)!(n)!}$ such vectors, and therefore this many new LWE samples one can generate.

**Results on Generated Samples.** Next, we show how SALSA performs when we combine different numbers of existing samples to create new ones for model training. We use the above method but do not allow the same sample to appear more than once in a given combination. We fix $K$, which is the number of samples used in each linear combination of reused samples. Then, we generate $K$ coefficients for the combined samples, where each $k_i$ is randomly chosen from $\{-1, 0, 1\}$. Finally, we randomly select $K$ samples from a pregenerated set of samples, and produce a new sample from their linear combination with the $k_i$ coefficients. These new samples follow error distribution with the standard deviation less than or equal to $\sqrt{K}\sigma$.

We experiment with different values of $K$, as well as different numbers of times we reuse a sample in linear combinations before discarding it. The $\log_2$ samples required for secret recovery for each $(K,$ times reused) setting are reported in Table 8. The first key result is that the secret is recovered in all experiments, confirming that the additional error introduced via sample combination does not disrupt model learning. Second, as expected, sample requirements decrease significantly as we increase both $K$ and times reused.

**Samples vs. Sigma.** We observe that the number of samples needed for secret recovery increases linearly with $\sigma$, see Figure 7.

Table 20: **N** vs **q**. Results reported are proportion of total secret bits recovered for various $N/q$ combinations. Green cells mean the secret was fully guessed, yellow cells all the 1 bits were correctly guessed during training, and red cells mean SALSA failed. Fixed parameters: $h = 3$, $base_{in} = base_{out} = 81$. 1/1 encoder layers, 1024/512 embedding dimension, 16/4 attention heads, 2/8 loops.

| $N$ | $\log_2(q)$ | | | | | | | | | |
|---|---|---|---|---|---|---|---|---|---|---|
| | 6 | 7 | 8 | 9 | 10 | 11 | 12 | 13 | 14 | 15 |
| **30** | 0.90 | 1.0 | 1.0 | 1.0 | 1.0 | 1.0 | 0.9 | 0.97 | 1.0 | 1.0 |
| **50** | 0.94 | 1.0 | 1.0 | 1.0 | 1.0 | 1.0 | 0.94 | 0.98 | 1.0 | 1.0 |
| **70** | 0.96 | 1.0 | 1.0 | 1.0 | 1.0 | 1.0 | 0.96 | 1.0 | 1.0 | 1.0 |
| **90** | 0.97 | 0.97 | 1.0 | 1.0 | 0.97 | 1.0 | 0.97 | 0.97 | 0.97 | 0.99 |

Table 21: **q** vs **d**. Results reported are proportion of total secret bits recovered for various $q/d$ combinations. Green cells mean the secret was fully guessed, yellow cells all the 1 bits were correctly guessed during training, and red cells mean SALSA failed. Fixed parameters: $N = 50$, $base_{in} = base_{out} = 81$. 1/1 encoder layers, 3040/1024 embedding dimension, 16/4 attention heads, 2/8 loops.

| **d** | $\log_2(q)$ | | | | | | | | | |
|---|---|---|---|---|---|---|---|---|---|---|
| | 6 | 7 | 8 | 9 | 10 | 11 | 12 | 13 | 14 | 15 |
| **.06** | 0.94 | 1.0 | 1.0 | 1.0 | 1.0 | 1.0 | 0.94 | 0.98 | 1.0 | 1.0 |
| **.08** | 0.92 | 0.92 | 1.0 | 0.92 | 0.94 | 0.92 | 0.94 | 0.94 | 0.94 | 0.94 |
| **.10** | 0.90 | 0.94 | 0.96 | 0.90 | 0.90 | 0.92 | 0.90 | 0.92 | 0.94 | 0.92 |

Table 22: **$B_{in}$** v. **$B_{out}$**. Effect of input and output integer base representation on $\log_2$ samples needed for secret recovery. In each row, the **bold** numbers represent the lowest $\log_2$ samples needed for this value of $B_{in}$. Fixed parameters: $n = 50$, hamming 3, 2/2 encoder layers, 1024/512 embedding dimension, 16/4 attention heads, and 2/8 loops.

| $B_{in}$ | $B_{out}$ | | | | |
|---|---|---|---|---|---|
| | 3 | 7 | 17 | 37 | 81 |
| **7** | 25.8 | **24.0** | 25.4 | 24.5 | 24.9 |
| **17** | - | 25.9 | 27.2 | 25.6 | **25.4** |
| **37** | 22.8 | **22.1** | 22.6 | 22.2 | 22.9 |
| **81** | 22.2 | **22.1** | 22.4 | 21.9 | **22.1** |

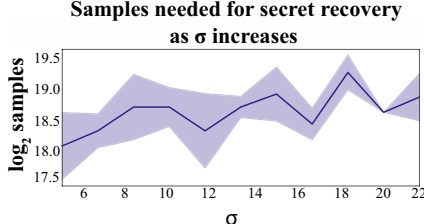

Figure 7: $log_2$ **samples vs. $\sigma$, fixed $n$.** As $\sigma$ increases, the $log_2$ samples required for a fixed $n = 50$, $h = 3$ increases linearly.

Figure 8: **Effect of Sample Reuse on Sample Efficiency.** Sample reuse via linear combinations greatly improves sample efficiency. The secret is recovered in all experiments, indicating that error introduced by sample combination does not degrade performance. Parameters: $n = 50$, Hamming 3, 2/2 encoder layers, 1024/512 embedding, 16/4 attention heads, 2/8 loops.

| **K** | **Times Samples Reused** | | | | |
|---|---|---|---|---|---|
| | 5 | 10 | 15 | 20 | 25 |
| **1** | 20.42 | 21.915 | 20.215 | 17.610 | 17.880 |
| **2** | 19.11 | 20.605 | 18.695 | 18.650 | 16.490 |
| **3** | 20.72 | 19.825 | 17.395 | 18.325 | 16.200 |
| **4** | 19.11 | 19.065 | 17.180 | 15.405 | 16.355 |