# OpenReview forum: "SALSA: Attacking Lattice Cryptography with Transformers"
_NeurIPS.cc/2022/Conference — NeurIPS 2022 Accept_

### Official Review · Reviewer_sMrM · 2022-06-27

**Rating:** 6
**Confidence:** 3
**Soundness:** 3 good
**Presentation:** 3 good
**Contribution:** 2 fair

**Summary:**

This paper tackles LWE, a so-called *hard learning problem* by training transformers models. The general idea is to feed a model with tuples of data/labels that are instances of an LWE distribution. If the model learns something, that means that it has implicitly guessed the secret variable $s$. The authors then proposes two ways to *extract* the secret $s$ from a model, in order to solve LWE.

After first demonstrating the efficiency of transformers models on modular arithmetic, the authors apply their technique on a special instance of LWE, namely with low-density binary secrets, in dimensionalities not yet close cryptographic-use, but still far from trivial. Based on these experimental outcomes, the authors argue to what extent the technique could scale to more realistic instances of LWE for cryptographic puposes.

**Questions:**

Any thought regarding my comment on the limitation of gradient descent? Do you think that it formally applies to your problem? Regardless of your answer, I think it would be valuable to add a discussion about that.


**Ethics Review Area:**

["I don’t know"]

**Limitations:**

No big limitation, except maybe the one I raised in the strength/weaknesses.

**Strengths And Weaknesses:**

Except for a few typos, the paper is well-written and easy to follow. The authors concisely introduce all the cryptographic material necessary to grasp the LWE problem. The principles of the attack are intuitive to follow. The parameters of the problem are clearly stated and discussed. The only drawback I could notice is that it does not introduce the transformers model, nor its variants that are yet mentioned and discussed throughout the paper. I guess that most of the target audience is familiar with such models, but I wish it could not have been assumed.
Overall, I enjoyed reading this paper. Moreover, the contribution seems original enough to me, and to the best of my knowledge, I am not aware of missing references close to this work.

My main concern about this papers covers the significance of the results. The authors deal with some parameters of the problem (the dimensionality $n$, the density of the secret, the modulus $q$ that are not yet cryptographic standards, although this is honestly and clearly stated by the authors. And despite, the parameters remain far from trivial.
Yet, the latter point sets the ground for the authors to argue to what extent their work could be extended to higher parameters in future works. Whether this extension is credible would imply the significance of the paper. And so far, I am not fully convinced that the difficulty only results from the intrinsic hardness of the LWE problem (as hypothesized l.272). I elaborate hereafter.

I have been aware of some works [1,2] explaining and quantifying the length of the initial plateau in the loss function in Figure 3. For example, it is known that the *learning parity* problem (i.e. for $q=2$, and $\sigma=0$), can be efficiently solved thanks to Gaussian elimination, but cannot be efficiently solved with gradient descent, regardless of the underlying model ($i.e.$ whether using transformers or not). Indeed, the initial plateau may have a length exponential with the parameter $n$ (removing the sparsity assumption). I assume that with low noise levels such as those considered in the paper, and with higher values of $q$, the same argument might apply. If so, this would be provably hard to solve (some instances of) LWE with gradient descent, whereas those LWE instances would not be cryptographically *hard*.

[1] https://proceedings.neurips.cc/paper/2020/hash/e7e8f8e5982b3298c8addedf6811d500-Abstract.html

[2] https://proceedings.mlr.press/v70/shalev-shwartz17a.html

---

> ### Author Response · Authors · 2022-08-01
> **Response to Reviewer sMrM**
>
> We thank the reviewer for this provocative question and helpful feedback.
>
> __Limitations of gradient descent for solving LWE__:  We have read the paper by Shalev-Schwart, Shamir and Shammah (SSSS henceforth), and we will add a discussion of it in the camera ready version. Meanwhile, we would like to make the following observations.
> 1. The LWE problem amounts to (discrete) linear regression on a $n$ dimensional torus with radius $q$. When $q$ is infinite, LWE reduces to pure linear regression, and gradient methods will succeed. For finite values of $q$ the gradients are informative, unless the model’s prediction for one coordinate is on the opposite side of the torus from the true solution (e.g. $q$=100, true value is 0, prediction is 50). At this point, the projection of the gradient along the coordinate axis is uninformative, since either direction on the torus points towards the true solution. For large $q$, this situation is very uncommon, and so gradient methods should work. For very small $q$, this will happen more often, and gradient methods will be perturbed. In the degenerate case $q$=2, gradient methods will always fail, as you observe.
> 	      For the $q$=251 used in SALSA, gradient methods can recover secrets (i.e. solve the modular linear regression problem) for $n$ up to 128 so far. For fixed values of $q$ and $\sigma$, we do not think the situation should be worse for larger $n$ or Hamming weight. In other words, we do not believe that, apart from the specific case of very small $q$ ($q$<10), LWE belongs to the “failing gradient” class of problems described in SSSS. We are inspired by your comments to test SALSA on larger $q$ values.
> 	     Interestingly, we see this intuition about the size of $q$ also in the classical lattice reduction approach to solving LWE. [Laine-Lauter](https://eprint.iacr.org/2015/176) use LLL to show concrete polynomial-time attacks against large $q$ and explore the boundary of how small $q$ can be before these attacks start to fail.  The intuition there is that when $q$ is large, LLL directly finds a vector which is “small enough” compared to $q$ to break the system.  When $q$ is small it cannot find a small enough vector.  This is further explored in [Chen-Chua-Lauter-Song](https://eprint.iacr.org/2020/539.pdf) which gives concrete running times for generalizations of LLL and finds the border for the size of $q$ where these attacks start to fail.
> 2.  While it is true that small $q$ reduces the information content of the gradient, this is aggravated in SSSS because they focus on SGD with batch size 1. By using larger batches (at least 128 examples), we average gradients over several examples, thus reducing the noise. The use of the Adam optimizer probably helps as well, since it averages each gradient step with previous steps. We had noticed, at the beginning of this research, that Adam outperformed simpler schemes like SGD, and we believe your comment provides the explanation. Thank you again. We will mention this in the paper.
> 3.  The Flat Activation problem mentioned in section 5 of SSSS was one of the main reasons why ReLU became popular. We only use ReLU activation. The layer normalization performed in the transformer also helps avoid vanishing gradients.
>
> __Comment about plateau being exponential in length $n$__: We find this comment interesting, but experimentally, we have observed that the plateau length appears to scale more with Hamming weight than dimension. For example, the plateau is much longer for a $n=30$, Hamming weight 5 secret than for an $n=110$, Hamming weight 3 secret. This reinforces our intuition that Hamming weight, not problem dimension, is the key obstacle to scaling SALSA (see Section 5.4).

---

### Official Review · Reviewer_WA3X · 2022-07-07

**Rating:** 7
**Confidence:** 3
**Soundness:** 3 good
**Presentation:** 3 good
**Contribution:** 3 good

**Summary:**

The paper performs cryptanalysis of the LWE problem using transformer models. They show that transformers can learn modular arithmetic. This justifies that, in principle, a transformer can learn the mapping from LWE inputs to outputs. They then show two ML-based cryptanalysis strategies which involve learning the LWE function and using this function to recover the secret. They evaluate on moderately-sized LWE problems.

**Questions:**

The strategy relies on being able to make predictions on these a_i vectors given random LWE samples. Could you attempt to measure the train/test overlap here? What is the closest vector in the (expanded) training set to the a_i vectors?

Does SALSA ever manage to extract a Hamming weight 5 secret? I see Hamming weight 4 secrets are possible if n < 70. Can you argue a bit more why this small Hamming weight limitation seen in the experiments is not inherent to the approach?

The training data expansion technique seems to bear some resemblance to sieving-based SVP algorithms. Could you comment on this? Could it be helping?

Since an epoch consists of 300000 examples (rather than a full pass through the training set), are there any examples seen multiple times?

**Limitations:**

I think the authors discuss limitations with SALSA as it exists, but I think they are more optimistic about SALSA than is backed up by their experiments.

**Strengths And Weaknesses:**

Strengths:

The paper is a pleasant read. It includes a good amount of problem-specific background.

SALSA is a novel attack on LWE. Regardless of whether SALSA in its current incarnation will be able to scale to larger problems, I think this is an interesting path for further investigation. SALSA also appears to have some principled justification.

There is a healthy amount of ablation and understanding how SALSA performs at different parameter settings. This is important for any new application of ML, and especially for a cryptanalysis technique.

Weaknesses:

The modular arithmetic results are not very well connected to the LWE results. My understanding of models' struggles with arithmetic is more that they're bad at extrapolating beyond the arithmetic they see in training. Modular arithmetic doesn't really require extrapolation, which might make this less interesting a result.

SALSA is not as interpretable as existing cryptanalysis techniques. That is, while it may be able to argue that a certain parameter setting is breakable, it is difficult to extrapolate how well SALSA will work on more real-world problems. Indeed, there is already some strange non-monotonicity in SALSA's performance: from n=90 -> n=110 (in Table 2), SALSA requires fewer samples and less running time.

It is unclear whether SALSA can simultaneously handle higher Hamming weights and higher dimension.

---

> ### Author Response · Authors · 2022-08-01
> **Response to Reviewer WA3X**
>
> Thank you for your insightful feedback. Our responses to your questions and comments are below.
>
> __Measuring the train/test overlap__: We randomly generate RLWE samples during training and testing (line 211). There are $n^q$ possible RLWE samples for a given problem setting. For an example to have a significant probability of appearing twice, we would need to generate on the order of $n^{q/2}$ examples (this is an instance of the [birthday attack](https://en.wikipedia.org/wiki/Birthday_attack)). Since our training and test sets never exceed a few billion examples, the probability of overlap is negligible.
>
> __Hamming weight 5 secret, scaling secret density__: SALSA has recovered Hamming weight 5 secrets for $n=30$, but these results were infrequent and thus not included in the paper. Since the submission, SALSA has also solved $n=80$, Hamming weight 4 problems. We agree that secret density is the main scaling challenge for SALSA, and we believe Table 4 provides strong intuition for how we can scale density. Table 4 shows that SALSA can recover secrets with much higher density when values in input $\mathbf{a}$ have restricted range (i.e. < $p \cdot q$). Our interpretation of this result is that higher density secrets are more difficult because they cause $b$ values to “wrap around” the modulus more times. When $\mathbf{a}$ values are restricted, the $b$ values “wrap” less for higher density secrets, enabling high density recoveries (Table 4). Since submitting this paper, we have been experimenting with methods that account for this wraparound behavior during learning, and have recovered secrets with higher Hamming weight (up to Hamming=10 for $n=20$).
>
> __Can SALSA handle higher Hamming weight and dimension simultaneously?__ Scaling $n$ and Hamming weight together is a key goal for future work based on SALSA. Our plan for scaling Hamming weight is outlined above, while our plan for scaling $n$ is mainly focused on increasing encoder/decoder dimensions. This is motivated by results in Tables 3, 16, and 17, which indicate that increasing the encoder/decoder dimension would allow SALSA to  handle higher dimensions. We do not think these two directions (changing the learning algorithm to account for modular arithmetic and scaling model architecture) are incompatible, and we are working to unify them.
>
> __Training examples seen multiple times?__ As mentioned previously, we randomly generate training examples, so the probability that the same example appears twice is negligible so long our training set has less than $n^{q/2}$ examples. As discussed in Section 6 (lines 305-317), we experiment with sample reuse, in which samples are reused up to K times before being discarded and find this significantly improves SALSA’s efficiency (Figure 4).
>
> __Relation to SVP sieving problem:__ Indeed, recovering the secret vector given many LWE samples amounts to recovering the shortest non-zero vector in a certain lattice, which we show can be done with machine learning via SALSA. Alternatively, in a well-known existing approach, the uSVP lattice attack, the key idea is to encode the information of the secret vector into the description of a lattice, and to use (exponential time) sieving algorithms to find the shortest vector in that lattice. With SALSA, we don’t need to sieve in the lattice to find the secret, we are using a different approach (model training plus designing a distinguisher which uses the trained model). So it’s unclear how the SVP sieving algorithm can play a role in SALSA, and we see it as a different way of solving the same problem. We would appreciate any further clarification to the question.
>
> __Modular arithmetic not well-connected to other results:__  We apologize if the presentation is confusing: Section 3 shows that we can achieve modular multiplication with high accuracy, which we need to make SALSA work.  However Section 3 also shows that in the multi-dimensional analogue, we do not directly achieve high accuracy.  So we are forced to come up with a different approach: designing a distinguisher which can use the half-trained (low accuracy) model for the multi-dimensional multiplication plus addition problem (i.e. inner product of vectors) to recover the secret.  The modular arithmetic results themselves are novel and so we presented them separately first, and they are also a core concept and a key ingredient for SALSA (multidimensional modular arithmetic).  Prior work on training ML to do modular arithmetic did not succeed [50,51], so we thought it important to showcase our experiments which successfully performed modular arithmetic using ML models. We can strengthen the connections between this part of the paper and SALSA in the camera ready version.
>
> __Non-monotonicity in Table 2:__ We believe the decrease in samples required and runtime for $n > 90$ vs. $n=90$ is due to the increased architecture size used for the $n > 90$ experiments (see caption of Table 2).

---

> > ### Comment · Reviewer_WA3X · 2022-08-08
> > **Thank you for the reply.**
> >
> > Thank you! I've read through the response and am happy with the comments.

---

### Official Review · Reviewer_bqbN · 2022-07-10

**Rating:** 3
**Confidence:** 4
**Soundness:** 2 fair
**Presentation:** 3 good
**Contribution:** 2 fair

**Summary:**

The paper proposes a machine learning approach to modular arithmetic and to breaking LWE-based cryptosystems based on said arithmetic.


**Questions:**

Do you agree that the running time of secret recovery roughly corresponds to the number of possible secrets?


**Limitations:**

yes

**Strengths And Weaknesses:**

The solved instances are trivial. For example, the last instance in Table 2 corresponds to finding a 128-bit secret of which all but three bits are zero. There are only $128 choose 3 = 341,376$ such secrets. I suspect this number of secrets can be tried in much less than the 23 hours it took this work. I think it's not appropriate to say this is comparable to the Darmstadt challenge where the secret is chosen uniformly at random from the whole domain, that is $n$ values modulo $q$, where $n \in [40,120]$ and $q$ is several thousand. The density of 0.002 is used with $n$ more than 10,000 leading to a much much larger number of possibilities.

The authors only show figures for reducing density with increasing dimension. This limits the possibilities for the secret. I think it's therefore overblown to say that the solution "may scale to attack real-world LWE-based cryptosystems".

---

> ### Author Response · Authors · 2022-08-01
> **Response to Reviewer bqbN**
>
> While we fully acknowledge that SALSA cannot yet attack real-world cryptographic implementations, our current results remain significant because they *open a new field of machine learning-based cryptanalysis of post-quantum cryptography (PQC) algorithms*. This is the first work to succeed in demonstrating an ML-based attack against lattice-based cryptosystems, specifically the underlying LWE hardness assumption. A major hurdle to overcome was showing that ML models can be trained to do modular arithmetic (see our Section 3), which is a trivial task for number theorists.
>
> Now that we have demonstrated the possibility of training ML models to do modular arithmetic (previously thought not possible in earlier work – see [50,51] in our paper), we use this as a building block to design and explore attacks on lattice-based systems. Given that LWE has recently been selected by NIST as its PQC standard, it is very important that the community investigate all angles of attack on this problem. We have shared our work in this direction so far with the NIST PQC team. It is widely acknowledged that cryptanalysis of LWE methods is a critical problem, and our method introduces a new approach which will open a new line of research in the area.
>
> __Runtime of secret recovery corresponds to the number of possible secrets?__
> The running time of SALSA does not depend on the Hamming weight = 3 (e.g. this is never given as a parameter), so you cannot make a direct runtime comparison to the number of possible secrets for any fixed Hamming weight.  We show a wide range of experiments, including one set achieving much higher Hamming weight in a restricted setting (see Table 4, with recoveries up to Hamming weight 15).  Interestingly, we observe that the number of samples used for training is relatively flat as dimension increases from dimension 30 to 128 (Table 2), which is radically different from the case for lattice reduction algorithms.  This in itself is motivation for further study in this direction.
>
> __Other points re: SALSA’s runtime and comparison to exhaustive search__:
> Overall, we do not have enough information yet to accurately quantify SALSA’s runtime.  Our current attack is largely unoptimized. We train a transformer from scratch for each attack, so many LWE samples are “wasted” in teaching the model what tokens to expect, etc. Pretraining transformers to use in our attacks may significantly reduce SALSA’s runtime and sample requirements. Additionally, we are conservative about memory use (e.g. use small models, only use a few GPUs per training run) which likely increases runtime. Overall, much more work is required before we feel confident making a definitive statement about SALSA’s secret recovery runtime and how it scales.
>
> Indeed, the exhaustive search approach you reference is always an option and may run faster than our current experiments for low Hamming weight. However, an exhaustive search requires knowing the exact hamming weight in advance, or iterating through all possible Hamming weights. Even though we have only succeeded unconditionally with low hamming weight examples so far, SALSA does not a priori know the secret Hamming weight.
>
> Additionally, our results suggest that SALSA’s difficulty with higher Hamming weights comes not from the increase in the dimension or the number of possible keys (as is the case for exhaustive search), but rather the fact that modular arithmetic becomes more difficult for models to learn as Hamming weight increases, as discussed in Section 5.4. Once we achieve desirable accuracy on testing samples, our secret recovery algorithms run in time *linear with respect to the dimension*, regardless of Hamming weight. In conclusion, we agree that Hamming weight is a limiting factor for SALSA which we are continuing to improve, but it doesn’t necessarily make the problem exponentially hard for SALSA, as it does in the case of exhaustive search.
>
> __Comparison to Darmstadt challenges__:
> In our text, we say that SALSA’s results are comparable to a version of the Darmstadt challenges with “sparse, binary secrets” (lines 369-370).  In other words, we do not compare SALSA’s performance to the Darmstadt challenges themselves. We will edit that sentence to clarify that we are attacking lattice dimensions in the same range as the Darmstadt challenges, but that we restrict to the easier case of sparse binary secrets. The point we are trying to make in referencing Darmstadt and other LWE-based homomorphic encryption (HE) methods is that the dimensions and the densities we consider are connected to real world problems, even though not when combined. If we were breaking any outstanding Darmstadt challenges or HE schemes, we would say so.

---

> > ### Comment · Reviewer_bqbN · 2022-08-08
> > **Baseline issues**
> >
> > Thank you for your remarks. Even if the Hamming weight isn't given to the algorithm, I think the results in Table 2 should be put in perspective by mentioning the number of possible secrets.
> >
> > Re Table 4, can the authors think of a baseline to compare the performance?

---

> > > ### Author Response · Authors · 2022-08-09
> > > **Response to baseline issues**
> > >
> > > We are happy to add the number of possible secrets as a baseline for Table 2.
> > >
> > > For Table 4, we believe the number of possible secrets for a given $n$/density would be the most appropriate baseline, since our experiments in Table 4 restrict the range of values in $\mathbf{a}$ but not the number of possible secrets. As a preview, for the first line of Table 4, the $log_2$ number of possible secrets is $29$ (50 choose 8), while for the last line, the $log_2$(secrets) is $41$ (50 choose 15). We appreciate this suggestion. Adding this baseline will further highlight the improvement in SALSA’s performance when $\mathbf{a}$ values are restricted and strengthen our argument that finding better ways of teaching modular arithmetic to models would enable SALSA to scale.

---

### Official Review · Reviewer_fdwD · 2022-07-10

**Rating:** 7
**Confidence:** 2
**Soundness:** 3 good
**Presentation:** 3 good
**Contribution:** 3 good

**Summary:**

- The authors propose SALSA, an attack on LWE-based cryptographic schemes, which has emerged as quantum-resistant cryptosystem, based on machine learning.
- The authors demonstrate that the process of modular arithmetic can be trained using transformer.
- Using trained transformer, the authors show that the instances from LWE can be distinguished from random instances.
- The authors propose two secret recovery algorithms, `direct secret recovery` and `distinguisher secret recovery`, and they justify these algorithms theoretically.

**Questions:**

- Does the runtime on Table 2 contain the training time? If not I want to see the training time of transformers used in the paper.
- If the authors can conduct other baseline methods on their machine, I want to see the comparison on runtime required for SALSA and the baseline methods on the same device.

POST REBUTTAL COMMENTS : The authors answer the questions. My concerns have been addressed and I decide to maintain my score.

**Limitations:**

That said, I wish that the authors include ethics or broader-impact statement in the paper or appendix.

**Strengths And Weaknesses:**

# Strengths
- The idea to perform modular arithmetic using a transformer seems interesting and novel.
- The method can recover secret even though the trained transformer has low-accuracy.
- Tables in the appendix are very helpful. For example, Table 20 and Table 21 helps me understand the broad effectiveness of the method.

# Weaknesses
- The algorithms require large computational resources and long runtime.
- Scalability of the method with respect to the dimension of the lattice seems poor.

---

> ### Author Response · Authors · 2022-08-01
> **Response to Reviewer fdwD**
>
> Thank you for your feedback. Our responses to your questions and comments are below.
>
> __Runtime in Table 2__: Table 2 contains the total time to secret recovery for each parameter setting. This includes the runtime of model training and secret recovery, which is run during the validation step of each training epoch.
>
> __Compare to baseline methods__: Our methods are so different from other LWE attacks that it would be a significant engineering effort to convert these to run on our cluster. Beyond this, in a head-to-head time comparison, existing lattice attacks would win for some of the parameter choices, as mentioned in Section 6 (for runtimes of other lattice attacks, see reference [20] in our paper). However, we are not claiming that SALSA outperforms existing methods. Instead, it represents a novel approach to a longstanding cryptographic problem, with promising initial results and the potential to scale.
>
> __Ethics and broader impact__: We can add a discussion of ethics and broader impact to Section 8 containing the following points. We will first discuss the value of our work in alerting the cryptographic and ML communities to the risk of ML-based attacks on PQC. Even if current attacks do not succeed, providing early warning of potential threats is an important contribution. Second, we will emphasize that SALSA represents a proof of concept that cannot be used against real-world implementations (i.e. the PQC schemes which NIST standardized on July 5) and note that additional scaling work will be necessary before these techniques will be relevant to attacking real-world cryptosystems. Finally, we will disclose that we sent a copy of our paper to the NIST PQC standardization group before submitting. In their response, they shared that they value this type of work, since it can indicate potential attack directions.

---

### Meta-Review · Area_Chair_wJH9 · 2022-08-24

**Recommendation:** Accept
**Confidence:** Less certain

**Metareview:**

The authors propose SALSA: a machine learning attack on cryptographic schemes based on Learning With Errors (LWE) as the underlying hard problem. They show that SALSA recovers secrets of small and medium size LWE instances with sparse binary secrets. The main selling point is that if the attack could scale up, it may pose a real treat to real-world LWE-based cryptosystems.

The reviewers found use of transformers to perform modular arithmetic interesting. At the same time, I agree with the reviewers that given the current computational complexity of the attack and given its poor scalability in the dimension of the lattice could potentially prevent its deployment in real-world systems. As such despite the merit of the paper, I find it to be a borderline submission.

**Award:**

No

---

### Decision · Program_Chairs · 2022-09-14

Accept